# Provably (More) Sample-Efficient Offline RL with Options

**Xiaoyan Hu**
Department of Computer Science and Engineering
The Chinese University of Hong Kong
Hong Kong SAR, China
`xyhu21@cse.cuhk.edu.hk`

**Ho-fung Leung**
Independent Researcher
Hong Kong SAR, China
`ho-fung.leung@outlook.com`

## Abstract

The options framework yields empirical success in long-horizon planning problems of reinforcement learning (RL). Recent works show that options improves the sample efficiency in *online* RL where the learner can actively explores the environment. However, these results are no longer applicable to scenarios where exploring the environment online is risky, e.g., automated driving and healthcare. In this paper, we provide the first analysis of the sample complexity for offline RL with options, where the agent learns from a dataset without further interaction with the environment. We propose the **PE**ssimistic **V**alue **I**teration for Learning with **O**ptions (PEVIO) algorithm and establish near-optimal suboptimality bounds (with respect to the novel information-theoretic lower bound for offline RL with options) for two popular data-collection procedures, where the first one collects state-option transitions and the second one collects state-action transitions. We show that compared to offline RL with actions, using options not only enjoys a faster finite-time convergence rate (to the optimal value) but also attains a better performance (when either the options are carefully designed or the offline data is limited). Based on these results, we analyze the pros and cons of the data-collection procedures, which may facilitate the selection in practice.

## 1 Introduction

Planning in long-horizon tasks is challenging in reinforcement learning (RL) (Co-Reyes et al., 2018; Eysenbach et al., 2019; Hoang et al., 2021). A line of study proposes to accelerate learning in these tasks using temporally-extended actions (Fikes et al., 1972; Sacerdoti, 1973; Drescher, 1991; Jiang et al., 2019; Nachum et al., 2019; Machado et al., 2021; Erraqabi et al., 2022). One powerful approach is the *options* framework introduced by Sutton et al. (1999), where the agent interacts with the environment with closed-loop policies called *options*. Empirical success (Tessler et al., 2017; Vezhnevets et al., 2017) shows that options help achieve sample-efficient performance in long-horizon planning problems.

To provide a theoretical guarantee to the options framework, recent works have focused on the sample complexity of RL with options in the *online* setting, where the agent continuously explores the environment and learns a *hierarchical policy* to select options. Brunskill and Li (2014) establish a PAC-like sample complexity of RL with options in the semi-Markov decision processes (SMDPs), where temporally-extended actions are treated as indivisible and unknown units. Later, Fruit and Lazaric (2017) provide the first regret analysis of RL with options under the Markov decision processes (MDPs) framework. While their proposed algorithm attains a sublinear regret, it requires prior knowledge of the environment, which is not usually available in practice. To address this problem, Fruit et al. (2017) propose an algorithm that does not require prior knowledge, yet achieves

37th Conference on Neural Information Processing Systems (NeurIPS 2023).

a near-optimal regret bound. However, these results are inapplicable to many real-world scenarios where online exploration is not allowed. For example, it has been argued that in healthcare (Gottesman et al., 2019) and automated driving (Shalev-Shwartz et al., 2016), learning in an online manner is risky and costly. In these scenarios, *offline* learning, where the agent learns a policy from a dataset, is preferred. We note that there is a line of studies on the sample complexity of offline RL with primitive actions only (i.e., without the use of options) (Levine et al., 2020; Fu et al., 2020; Rashidinejad et al., 2021). Unfortunately, to the best of our knowledge, there have been no results reported on the offline RL with options.

In this paper, we make the following contributions. First, we derive a novel information-theoretic lower bound, which generalizes the one for offline learning with actions. Second, we propose the **PE**ssimistic **V**alue **I**teration for Learning with **O**ptions (PEVIO) algorithm and derive near-optimal suboptimality bounds for two popular data-collection procedures, where the first one collects state-option transitions and the second one collects state-action transitions. More importantly, we show that options facilitate more sample-efficient learning in both the finite-time convergence rate and actual performance. To shed light on offline RL with options in practice, we discuss the pros and cons of both data-collection procedures based on our analysis.

## 2 Related Work

**Learning with Options** Building upon the theory of *semi-Markov decision processes* (SMDPs) (Bradtke and Duff, 1994; Mahadevan et al., 1997), Sutton et al. (1999) propose to learn with options. Following their seminal work, learning with options has been widely studied in the function approximation setting (Sorg and Singh, 2010) and hierarchical RL (Igl et al., 2020; Klissarov and Precup, 2021; Wulfmeier et al., 2021). Discovering useful options has also been the subject of extensive research (Stolle and Precup, 2002; Riemer et al., 2018; Mankowitz et al., 2018; Harb et al., 2018; Hiraoka et al., 2019; Bagaria et al., 2021). Despite its empirical success, there have been fairly limited studies on the sample efficiency of learning with options. Brunskill and Li (2014) analyze the sample complexity bound for an RMAX-like algorithm for SMDPs. Fruit and Lazaric (2017) derive the first regret analysis of learning with options. They propose an algorithm that attains sublinear regret in the infinite-horizon average-reward MDP while requiring prior knowledge of the environment. Later, Fruit et al. (2017) remove this requirement.

**Offline RL** In the offline setting, a dataset that is collected by executing a *behavior policy* in the environment is provided, and the agent is asked to learn a near-optimal policy using only this dataset. A key challenge in offline RL is the insufficient coverage of the dataset (Wang et al., 2021), which is also known as distributional shift (Chen and Jiang, 2019; Levine et al., 2020). To address this problem, the previous study on sample-efficient learning assumes uniform coverage of the dataset (Liu et al., 2018; Chen and Jiang, 2019; Jiang and Huang, 2020; Yang et al., 2020; Xie and Jiang, 2020; Uehara et al., 2020; Qu and Wierman, 2020; Yin et al., 2021). This assumption is relaxed in recent works by pessimism principle (Xie et al., 2021; Rashidinejad et al., 2021; Jin et al., 2021).

## 3 Preliminaries

### 3.1 Episodic MDP with Options

Let $\Delta(\mathcal{X})$ denote the probability simplex on space $\mathcal{X}$ and $[N] := \{1, \cdots, N\}$ for any positive integer $N$. An episodic MDP with options is a sextuple $\mathcal{M} = (\mathcal{S}, \mathcal{A}, \mathcal{O}, H, \mathcal{P}, r)$, where $\mathcal{S}$ is the state space, $\mathcal{A}$ the (primitive) action set, $\mathcal{O}$ the finite set of options, $H$ the length of each episode, $\mathcal{P} = \{P_h : \mathcal{S} \times \mathcal{A} \mapsto \Delta(\mathcal{S})\}_{h \in [H]}$ the transition kernel, $r = \{r_h : \mathcal{S} \times \mathcal{A} \mapsto [0, 1]\}_{h \in [H]}$ the deterministic reward function.[1] We define $S := |\mathcal{S}|$, $A := |\mathcal{A}|$, and $O := |\mathcal{O}|$. A (Markov) *option* $o \in \mathcal{O}$ is a pair $(\pi^o, \beta^o)$ where $\pi^o = \{\pi_h^o : \mathcal{S} \mapsto \Delta(\mathcal{A})\}_{h \in [H]}$ is the option's policy and $\beta^o = \{\beta_h^o : \mathcal{S} \mapsto [0, 1]\}_{h \in [H]}$ is the probability of the option's *termination*. For convenience, we define $\beta_{H+1}^o(s) = 1$ for all $(s, o) \in \mathcal{S} \times \mathcal{O}$, i.e., any option is terminated after the end of an

---

[1]Our results can be directly generalized to stochastic rewards.

episode. We assume that the initial state $s_1$ is *fixed*.[2] Upon arriving at state $s_h$ at any timestep $h \in [H]$, if $h = 1$ (at the beginning of an episode), the agent selects option $o_1 \sim \mu_1(\cdot|s_1)$, where $\mu = \{\mu_h : \mathcal{S} \mapsto \Delta(\mathcal{O})\}_{h \in [H]}$ is a *hierarchical policy* to select an option at each state. Otherwise ($h \geq 2$), the agent first terminates option $o_{h-1}$ with probability $\beta_h^{o_{h-1}}(s_h)$. If option $o_{h-1}$ is terminated, she then selects a new option $o_h \sim \mu_h(\cdot|s_h)$ according to the hierarchical policy $\mu$. If option $o_{h-1}$ is not terminated, the agent continues to *use* option $o_{h-1}$ at timestep $h$, i.e., $o_h = o_{h-1}$. After that, the agent takes action $a_h \sim \pi_h^{o_h}(\cdot|s_h)$, receives a reward $r_h := r_h(s_h, a_h)$, and transits to the next state $s_{h+1} \sim P_h(\cdot|s_h, a_h)$. An episode terminates at timestep $H + 1$. A special case is that an action $a$ is an option $o$, such that $\pi_h^o(a|s) = 1$ and $\beta_h^o(s) = 1$ for any $(h, s) \in [H] \times \mathcal{S}$. For convenience, we use the notation $\mathcal{O} = \mathcal{A}$ to represent that each option corresponds to an action, which is the case in RL with primitive actions.

To define the $Q$-function and the value function, we introduce some useful notations.[3] Let $T = \{T_h : \mathcal{S} \times \mathcal{O} \mapsto \Delta(\mathcal{S} \times [H - h + 1])\}_{h \in [H]}$ and $U = \{U_h : \mathcal{S} \times \mathcal{O} \mapsto [0, H]\}_{h \in [H]}$ denote the *option transition function* and the *option utility function*, respectively. Particularly, for any $(h, s, o) \in [H] \times \mathcal{S} \times \mathcal{O}$, the option transition function $T_h(s'|s, o, \tau)$ is the probability that the agent uses option $o$ at state $s$ at timestep $h$, reaches state $s'$ at timestep $h + \tau$ without terminating option $o$ in these $\tau$ timesteps, and finally terminates option $o$ at state $s'$ at timestep $h + \tau$. The option utility function $U_h(s, o)$ is the expected cumulative reward within timesteps that the option is used without being terminated. Given any arbitrary series of functions $\{y_h : \mathcal{S} \mapsto \mathbb{R}\}_{h \in [H]}$, define the operator $[T_h y_{h+\tau}](s, o) := \sum_{s' \in \mathcal{S}} T_h(s'|s, o, \tau) y_{h+\tau}(s')$ for any $(s, o, \tau) \in \mathcal{S} \times \mathcal{O} \times [H - h + 1]$. In the following, we derive the $Q$-function and the value function for learning with options. (The detailed proof can be found in Appendix B.)

**Theorem 1** ($Q$-function and value function). *For any hierarchical policy $\mu$ and $(h, s, o) \in [H] \times \mathcal{S} \times \mathcal{O}$, the $Q$-function is given by*

$$Q_h^\mu(s, o) := \mathbb{E}_\mu \left[ \sum_{h'=h}^H r_{h'}(s_{h'}, a_{h'}) \middle| s_h = s, o_h = o \right] = U_h(s, o) + \sum_{\tau \in [H-h+1]} [T_h V_{h+\tau}^\mu](s, o) \tag{1}$$

*and the value function is given by*

$$V_h^\mu(s) := \mathbb{E}_\mu \left[ \sum_{h'=h}^H r_{h'}(s_{h'}, a_{h'}) \middle| s_h = s, o_h \sim \mu_h(\cdot|s_h) \right] = \sum_{o \in \mathcal{O}} \mu_h(o|s) Q_h^\mu(s, o) \tag{2}$$

*where $V_{H+1}^\mu(s) = Q_{H+1}^\mu(s, o) = 0$ for any $(s, o) \in \mathcal{S} \times \mathcal{O}$.*

Intuitively, the first term $U_h(s, o)$ of the $Q$-function is the expected reward within timesteps that option $o$ is used without being terminated, and the second term $\sum_{\tau \in [H-h+1]} [T_h V_{h+\tau}^\mu](s, o)$ corresponds to the expected reward within timesteps after option $o$ is terminated and a new option is selected according to $\mu$. It can be shown that there exists an optimal (and deterministic) hierarchical policy $\mu^* = \{\mu_h^* : \mathcal{S} \mapsto \mathcal{O}\}_{h \in [H]}$ that attains the optimal value function, i.e., $V_h^*(s) = \sup_\mu V_h^\mu(s)$ for all $(h, s) \in [H] \times \mathcal{S}$ (Sutton et al., 1999).

## 3.2 Offline RL with Options

We consider learning with options in the offline setting. That is, given a dataset $\mathcal{D}$ that is collected by an experimenter through interacting with the environment, the algorithm outputs a hierarchical policy $\widehat{\mu}$. The sample complexity is measured by the *suboptimality*, i.e., the shortfall in the value function of the hierarchical policy $\widehat{\mu}$ compared to that of the optimal hierarchical policy $\mu^*$, which is given by

$$\text{SubOpt}_{\mathcal{D}}(\widehat{\mu}, s_1) := V_1^*(s_1) - V_1^{\widehat{\mu}}(s_1) \tag{3}$$

To derive a novel information-theoretic lower bound of $\text{SubOpt}_{\mathcal{D}}$, we first define some useful notations. For any hierarchical policy $\mu$, we denote by $\theta^\mu = \{\theta_h^\mu : \mathcal{S} \times \mathcal{O} \mapsto [0, 1]\}_{h \in [H]}$ its *state-option*

---

[2]Note that any $H$-length episodic MDP with a stochastic initial state is equivalent to an $(H + 1)$-length MDP with a fixed initial state $s_0$.

[3]The formal definitions can be found in Appendix A.

*occupancy measure*. That is, $\theta_h^\mu(s, o)$ is the probability that the agent selects a particular option $o$ at state $s$ at timestep $h$ (either when $h = 1$ or when the option $o_{h-1}$ used at the timestep $h - 1$ is terminated) when following the hierarchical policy $\mu$. With a slight abuse of the notation, we denote by $\theta_h^\mu(s) := \sum_{o \in \mathcal{O}} \theta_h^\mu(s, o)$ the *state occupancy measure* for any $(h, s) \in [H] \times \mathcal{S}$. Further, we define

$$Z_{\mathcal{O}}^\mu := \sum_{h,s} \theta_h^\mu(s), \ \overline{Z}_{\mathcal{O}}^\mu := \sum_{h,s,o} \mathbb{I}[\theta_h^\mu(s, o) > 0] \tag{4}$$

where $\mathbb{I}[\cdot]$ is the indicator function. Intuitively, $Z_{\mathcal{O}}^\mu$ is the expected number of timesteps to alternate a new option and $\overline{Z}_{\mathcal{O}}^\mu$ is the maximal number of state-option pairs that can be visited, when following the hierarchical policy $\mu$. The following proposition shows that options facilitate temporal abstraction and reduction of the state space.

**Proposition 1.** *For any hierarchical policy $\mu$, we have that $Z_{\mathcal{O}}^\mu \leq H$. If $\mu$ is deterministic, i.e., $\mu = \{\mu_h : \mathcal{S} \mapsto \mathcal{O}\}_{h \in [H]}$, we further have that $\overline{Z}_{\mathcal{O}}^\mu \leq HS$. All the above equalities hold when $\mathcal{O} = \mathcal{A}$.*

Next, we derive a novel information-theoretic lower bound of $\text{SubOpt}_{\mathcal{D}}$. The detailed proof can be found in Appendix C.

**Theorem 2** (Information-theoretic lower bound). *Let $\rho = \{\rho_h : \mathcal{S} \mapsto \Delta(\mathcal{O})\}_{h \in [H]}$ denote any hierarchical behavior policy to collect the dataset. Define the class of problem instances*

$$\mathcal{M}(C^{\text{option}}, z^*, \overline{z}^*) := \left\{ (M, \rho) : \text{ Exists deterministic } \mu^* \text{ of an episodic MDP } M \right.$$
$$\left. \text{ such that } \max_{h,s,o} \frac{\theta_h^{\mu^*}(s, o)}{\theta_h^\rho(s, o)} \leq C^{\text{option}}, Z_{\mathcal{O}}^{\mu^*} \leq z^*, \overline{Z}_{\mathcal{O}}^{\mu^*} \leq \overline{z}^* \right\}.$$

*Suppose that $C^{\text{option}} \geq 2$, $z^* \geq 1$, and $\overline{z}^* \geq \lfloor z^* \rfloor S$, where $\lfloor x \rfloor := \max\{n \in \mathbb{N} : n \leq x\}$ is the largest integer no greater than $x \in \mathbb{R}$. Then, there exists an absolute constant $c_0$ such that for any offline algorithm that outputs a hierarchical policy $\widehat{\mu}$, if the number of episodes*

$$K \leq \frac{c_0 \cdot C^{\text{option}} H z^* \overline{z}^*}{\epsilon^2}$$

*then there exists a problem instance $(M, \rho) \in \mathcal{M}(C^{\text{option}}, z^*, \overline{z}^*)$ on which the hierarchical policy $\widehat{\mu}$ suffers from $\epsilon$-suboptimality, that is,*

$$\mathbb{E}_M[\text{SubOpt}_{\mathcal{D}_1}(\hat{\mu}, s)] \geq \epsilon$$

*where the expectation $\mathbb{E}_M$ is with respect to the randomness during the execution of $\rho$ within MDP $M$.*

Theorem 2 shows that, when dataset $\mathcal{D}$ sufficiently covers the trajectories induced by $\mu^*$, i.e., $\max_{h,s,o} \theta_h^{\mu^*}(s, o)/\theta_h^\rho(s, o) \leq C^{\text{option}}$, at least $\Omega(C^{\text{option}} H Z_{\mathcal{O}}^* \overline{Z}_{\mathcal{O}}^*/\epsilon^2)$ episodes are required to learn an $\epsilon$-optimal hierarchical policy from dataset $\mathcal{D}$. Note that when $\mathcal{O} = \mathcal{A}$, it recovers the lower bound $\Omega(H^3 S C^*/\epsilon^2)$ for offline RL with primitive actions, where $C^*$ is the concentrability defined therein.

## 4 The PEVIO Algorithm

Inspired by the Pessimistic Value Iteration (PEVI) algorithm (Jin et al., 2021), we propose the **PE**ssimistic **V**alue **I**teration for Learning with **O**ptions (PEVIO) in Algorithm 1. Given a dataset $\mathcal{D}$ and the corresponding **O**ffline **O**ption **E**valuation (OOE) subroutine, whose details are specified in Sections 5.1 and 5.2, PEVIO outputs a hierarchical policy $\widehat{\mu} = \{\widehat{\mu}_h : \mathcal{S} \mapsto \Delta(\mathcal{O})\}_{h \in [H]}$.

To estimate the $Q$-function, given a dataset $\mathcal{D}$, PEVIO first constructs $(\widehat{T}, \widehat{U}, \Gamma)$ by the OOE subroutine (line 3). Specifically, $\widehat{T}_h$ and $\widehat{U}_h$ are the empirical counterparts of $T_h$ and $U_h$ presented in the $Q$-function given by Equation (1), respectively. In addition, $\Gamma$ is a penalty function computed based on dataset $\mathcal{D}$. We remark that the OOE subroutine varies when different data-collecting procedures

---

**Algorithm 1 PEssimistic Value Iteration for Learning with Options (PEVIO)**

---

1: Input: Dataset $\mathcal{D}$ and the corresponding **O**ffline **O**ption **E**valuation (OOE) subroutine.
2: Initialize: $\widehat{Q}_h(s, o) \leftarrow 0, \widehat{V}_h(s) \leftarrow 0, \widehat{V}_{H+1}(s) \leftarrow 0$ for any $(h, s, o) \in [H] \times \mathcal{S} \times \mathcal{O}$.
3: $(\widehat{T}, \widehat{U}, \Gamma) \leftarrow \text{OOE}(\mathcal{D})$.
4: **for** $h = H, H-1, \cdots, 1$ **do**
5:     **for** $(s, o) \in \mathcal{S} \times \mathcal{O}$ **do**
6:         $\overline{Q}_h(s, o) \leftarrow \widehat{U}_h(s, o) + \sum_{\tau=1}^{H-h+1} [\widehat{T}_h \widehat{V}_{h+\tau}](s, o) - \Gamma_h(s, o)$.
7:         $\widehat{Q}_h(s, o) \leftarrow \max\{0, \min\{\overline{Q}_h(s, o), H - h + 1\}\}$.
8:     **end for**
9:     **for** $s \in \mathcal{S}$ **do**
10:        $\widehat{\mu}_h(\cdot|s) \leftarrow \arg\max_{\mu_h}\langle\widehat{Q}_h(s, \cdot), \mu_h(\cdot|s)\rangle_{\mathcal{O}}$.
11:        $\widehat{V}_h(s) \leftarrow \langle\widehat{Q}_h(s, \cdot), \widehat{\mu}_h(\cdot|s)\rangle_{\mathcal{O}}$.
12:     **end for**
13: **end for**
14: Output: $\widehat{\mu} = \{\widehat{\mu}_h\}_{h\in[H]}$.

---

are considered and we provide the details in Sections 5.1 and 5.2, respectively. Given $\widehat{U}_h$, $\widehat{T}_h$, and $\Gamma_h$, the estimated $Q$-function $\widehat{Q}_h$ is the derived (lines 6 and 7). Particularly, $\overline{Q}_h$ computed in line 6 can be seen as first replacing $U_h$ and $T_h$ with their empirical counterparts $\widehat{U}_h$, $\widehat{T}_h$ in Equation (1), and then subtracting the penalty function $\Gamma_h$. Further, a hierarchical policy $\widehat{\mu}_h$ is constructed greedily with $\widehat{Q}_h$ (line 10), where $\langle f(\cdot), g(\cdot)\rangle_{\mathcal{O}} := \sum_{o\in\mathcal{O}} f(o)g(o)$ for any arbitrary functions $f, g$ defined on $\mathcal{O}$. Finally, given $\widehat{Q}_h$ and $\widehat{\mu}_h$, the corresponding estimated value function $\widehat{V}_h$ is computed (line 11). To analyze the suboptimality of the hierarchical policy $\widehat{\mu}$ output from PEVIO, we first provide the the following definition, which motivates the design of the penalty function $\Gamma$.

**Definition 1** ($\xi$-uncertainty quantifier for dataset $\mathcal{D}$). The penalty function $\Gamma = \{\Gamma_h : \mathcal{S} \times \mathcal{O} \mapsto \mathbb{R}^+\}_{h\in[H]}$ output from the OOE subroutine in Algorithm 1 (line 3) is said to be a $\xi$-uncertainty quantifier with respect to $\mathbb{P}_{\mathcal{D}}$ if the following event

$$\mathcal{E} = \{|\widehat{U}_h(s, o) - U_h(s, o) + \sum_{\tau=1}^{H-h+1} [(\widehat{T}_h - T_h)\widehat{V}_{h+\tau}](s, o)| \tag{5}$$
$$\leq \Gamma_h(s, o) \text{ for all } (h, s, o) \in [H] \times \mathcal{S} \times \mathcal{O}\}$$

satisfies that $\mathbb{P}_{\mathcal{D}}(\mathcal{E}) \geq 1 - \xi$, where $\mathbb{P}_{\mathcal{D}}$ is the joint distribution of the data collecting process.

In other words, the penalty function $\Gamma$ is a $\xi$-uncertainty quantifier if it upper bounds the estimation errors in the empirical option transition function $\widehat{T}$ and the empirical option utility function $\widehat{U}$. Next, we show that the suboptimality of $\widehat{\mu}$ output from PEVIO is upper bounded if $\Gamma$ is a $\xi$-uncertainty quantifier. (The detailed proof can be found in Appendix D.)

**Theorem 3** (Suboptimality of learning with options using dataset $\mathcal{D}$). *Let $\widehat{\mu}$ denote the hierarchical policy output by Algorithm 1. Suppose that $\Gamma = \{\Gamma_h\}_{h\in[H]}$ output from the OOE subroutine is a $\xi$-uncertainty quantifier. Conditioned on the successful event $\mathcal{E}$ defined in Equation (5), which satisfies that $\mathbb{P}_{\mathcal{D}}(\mathcal{E}) \geq 1 - \xi$, we have that*

$$\text{SubOpt}_{\mathcal{D}}(\widehat{\mu}, s_1) \leq 2 \sum_{h=1}^{H} \mathbb{E}_{\mu^*}[\Gamma_h(s_h, o_h)|s_1] \tag{6}$$

*where $\mathbb{E}_{\mu^*}[g(s_h, o_h)] = \sum_{(s,o)} \theta_h^{\mu^*}(s, o)g(s, o)$ for any $h \in [H]$ and arbitrary function $g : \mathcal{S} \times \mathcal{O} \mapsto \mathbb{R}$.*

*Remark* 1. Since the temporal structure of learning with options is much more complex than learning with actions, PEVIO is significantly different from the algorithms proposed for offline RL with primitive actions, such as PEVI (Jin et al., 2021) or VI-LCB (Xie et al., 2021), despite sharing a similar intuition. First, in terms of the algorithm design, PEVI and VI-LCB estimate (one-step) transition kernel and reward function to compute the $Q$-function of a state-action pair. However, by Equation (1), the $Q$-function of a state-option pair depends on multi-step transitions and rewards.

Hence, it is challenging to design the OOE subroutine and analyze the estimated $(\widehat{T}, \widehat{U}, \Gamma)$ for options. Indeed, if the dataset contains $(s, o, u)$ (state-option-utility) tuples, then $(T, U, \Gamma)$ can be estimated similarly to the case of learning with primitive actions. However, if the dataset contains only $(s, a, r)$ (state-action-reward) tuples, then it remains elusive to estimate and analyze $(T, U, \Gamma)$. Second, in terms of the suboptimality analysis, previous works on offline RL with primitive actions rely on the extended value difference lemma (Cai et al., 2020, Lemma 4.2), which also depends on the one-step temporal structure of actions and cannot be directly applied to our setting. Hence, to derive Theorem 3, it is non-trivial to generalize the extended value difference lemma to the options framework (See Lemma 8 in the Appendices).

## 5 Data-Collection and Suboptimality Analysis

In this section, we consider two data-collection procedures that are widely deployed in the options literature. The first one collects state-option-utility tuples (dataset $\mathcal{D}_1$) and similar datasets are utilized in the work of Zhang et al. (2023). The second one collects state-action-reward tuples (dataset $\mathcal{D}_2$) and is studied in a line of works (Ajay et al., 2021; Villecroze et al., 2022; Salter et al., 2022). Intuitively, dataset $\mathcal{D}_1$ requires smaller storage and enables efficient evaluation of the options, while dataset $\mathcal{D}_2$ provides richer information on the environment and even facilitates the evaluation of new options. For each dataset, we design the corresponding OOE subroutine and derive a suboptimality bound for the PEVIO algorithm. Based on these results, we further discuss the advantages and the disadvantages of both data-collection procedures, which sheds light on offline RL with options in practice.

### 5.1 Learning from State-Option Transitions

We consider dataset $\mathcal{D}_1 := \{(s_{t_i^k}^k, o_{t_i^k}^k, u_{t_i^k}^k)\}_{i \in [j^k], k \in [K]}$ consisting of state-option-utility tuples, which is collected by the experimenter's interaction with the environment for $K$ episodes using a hierarchical behavior policy $\rho = \{\rho_h : \mathcal{S} \mapsto \Delta(\mathcal{O})\}_{h \in [H]}$. More precisely, at timestep $t_i^k$ of the $k$th episode, the experimenter selects a new option $o_{t_i^k}^k$, uses it for $(t_{i+1}^k - t_i^k)$ timesteps, collects a cumulative reward of $u_{t_i^k}^k$ within these $(t_{i+1}^k - t_i^k)$ timesteps, and finally terminates this option at state $s_{t_{i+1}^k}^k$ at timestep $t_{i+1}^k$. For convenience, we define $t_{j^k+1}^k := H + 1$ for any $k \in [K]$.

Let $a \vee b := \max\{a, b\}$ for any pair of integers $a, b \in \mathbb{N}$. When dataset $\mathcal{D}_1$ is available, the OOE subroutine in Algorithm 1 is given by Subroutine 2. Particularly, Subroutine 2 incorporates the data splitting technique (Xie et al., 2021) (line 2). That is, given dataset $\mathcal{D}_1$, the algorithm randomly splits it into $H$ subdatasets $\{\mathcal{D}_{1,h}\}_{h \in [H]}$. Then $\widehat{T}_h$ and $\widehat{U}_h$ are constructed using subdataset $\mathcal{D}_{1,h}$ (lines 4-7).

To derive the suboptimality for $\widehat{\mu}$ output from PEVIO, we follow the previous study and make a standard assumption on the coverage of dataset $\mathcal{D}_1$.

**Assumption 1** (Single hierarchical policy concentrability for dataset $\mathcal{D}_1$). The experimenter collects dataset $\mathcal{D}_1$ by following a hierarchical behavior policy $\rho = \{\rho_h : \mathcal{S} \mapsto \Delta(\mathcal{O})\}_{h \in [H]}$. There exists some deterministic optimal hierarchical policy $\mu^*$ such that

$$C_1^{\text{option}} := \max_{h,s,o} \frac{\theta_h^{\mu^*}(s, o)}{\theta_h^{\rho}(s, o)} \tag{7}$$

(with the convention $0/0 = 0$) is finite.

In other words, Assumption 1 states that dataset $\mathcal{D}_1$ sufficiently covers the trajectories of state-option-utility tuples induced by some deterministic optimal hierarchical policy $\mu^*$. We derive an upper bound of $\text{SubOpt}_{\mathcal{D}_1}$ in the following theorem. (The detailed proof can be found in Appendix E.)

**Theorem 4** (Suboptimality for dataset $\mathcal{D}_1$). *Under Assumption 1, with probability at least $1 - \xi$, we have that*

$$\text{SubOpt}_{\mathcal{D}_1}(\widehat{\mu}, s_1) \leq \tilde{O}\left(\sqrt{\frac{C_1^{\text{option}} H^3 Z_{\mathcal{O}}^* \overline{Z}_{\mathcal{O}}^*}{K}}\right) \tag{8}$$

---

**Subroutine 2** **O**ffline **O**ption **E**valuation (OOE) for Dataset $\mathcal{D}_1$

---

1: **Input:** Dataset $\mathcal{D}_1 = \{(s_{t_i^k}^k, o_{t_i^k}^k, u_{t_i^k}^k)\}_{i \in [j^k], k \in [K]}$.

2: **Initialize:** Randomly split the dataset $\mathcal{D}$ into $H$ subdatasets $\{\mathcal{D}_{1,h}\}_{h \in [H]}$ with $|\mathcal{D}_{1,h}| = K/H$. More precisely, let $l := \{l_h\}_{h \in [H]}$ be a random partition of the set $[K]$, where $l_h := \{l_{h,j}\}_{j \in [K/H]} \subset [K]$ is uniformly sampled from $[K]$ such that $\cup_{h \in [H]} l_h = [K]$ and $l_h \cap l_{h'} = \emptyset$ for any $h \neq h'$. Then we have that $\mathcal{D}_{1,h} = \{(s_{t_i^k}^k, o_{t_i^k}^k, u_{t_i^k}^k)\}_{i \in [j^k], k \in l_h}$ for any $h \in [H]$. Let $n_h(s, o) := \sum_{k \in l_h} \mathbb{I}[h \in \{t_i^k\}_{i \in [j^k]}, s_h^k = s, o_h^k = o]$ denote the number of times that the experimenter selects a particular option $o$ at state $s$ at timestep $h$ in subdataset $\mathcal{D}_{1,h}$.

3: **for** $(h, s, o) \in [H] \times \mathcal{S} \times \mathcal{O}$ **do**

4:      **for** $(s', \tau) \in \mathcal{S} \times [H - h + 1]$ **do**

5:          $\widehat{T}_h(s'|s, o, \tau) \leftarrow \frac{\sum_{k \in l_h} \mathbb{I}[h \in \{t_i^k\}_{i \in [j^k]}, s_h^k = s, o_h^k = o, s_{h+\tau}^k = s']}{1 \vee n_h(s, o)}$

6:      **end for**

7:      $\widehat{U}_h(s, o) \leftarrow \frac{\sum_{k \in l_h} \mathbb{I}[h \in \{t_i^k\}_{i \in [j^k]}, s_h^k = s, o_h^k = o] u_h^k}{1 \vee n_h(s, o)}$

8:      $\Gamma_h(s, o) \leftarrow \tilde{O}\left(\sqrt{\frac{H^2}{n_h(s, o) \vee 1}}\right)$

9: **end for**

10: **Output:** $(\widehat{T} = \{\widehat{T}_h\}_{h \in [H]}, \widehat{U} = \{\widehat{U}_h\}_{h \in [H]}, \Gamma = \{\Gamma_h\}_{h \in [H]})$.

---

where $Z_{\mathcal{O}}^* := Z_{\mathcal{O}}^{\mu^*}$ and $\overline{Z}_{\mathcal{O}}^* := \overline{Z}_{\mathcal{O}}^{\mu^*}$.

Compared to the lower bound in Theorem 2, suboptimality bound (8) is near-optimal except for an extra factor of $H$.[4] More importantly, it shows that learning with options enjoys a faster convergence rate to the optimal value than learning with primitive actions. Recall that the VI-LCB algorithm (Xie et al., 2021) that learns with primitive actions attains the suboptimality bound $\tilde{O}(\sqrt{H^5 SC^*/K})$, where $C^*$ is the concentrability defined therein. When ignoring the concentrability parameters, the suboptimality bound (8) is smaller since $Z_{\mathcal{O}}^* \leq H$ and $\overline{Z}_{\mathcal{O}}^* \leq HS$.

*Remark* 2. While, in the worst case, both $Z_{\mathcal{O}}^*$ and $\overline{Z}_{\mathcal{O}}^*$ can scale with $H$ and $HS$, respectively, we note that in many long-horizon planning problems, they often scale with the *number of sub-tasks*, which are greatly smaller, especially for tasks that enables temporal abstraction and the reduction of the state space. For example, while the route-planning task of going from City A to City B by transportation takes thousands of primitive actions to finish, it can be efficiently solved by decomposing into the following sub-tasks: (1) going to the airport/train station in City A; (2) taking transportation to City B; and (3) reaching the final destination in City B, for which options are designed. In this case, both $Z_{\mathcal{O}}^*$ and $\overline{Z}_{\mathcal{O}}^*/S$ may only scale as $o(H)$. In other words, options facilitate more sample-efficient learning through temporal abstraction, i.e., sticking to an option until a sub-task is finished. Another concrete example is solving a maze, where options are often designed to move agents to bottleneck states (Şimşek and Barto, 2004; Solway et al., 2014; Machado et al., 2017) that connect different densely connected regions of the state space, e.g., doorways. In this case, while the number of option switches may grow proportionally to $H$, i.e., $Z_{\mathcal{O}}^*/H = O(1)$, the number of states to switch options can be greatly smaller than $S$, i.e., $\overline{Z}_{\mathcal{O}}^*/H = o(s)$. That is to say, options help improve the sample complexity by the reduction of the state space.

Further, we show that learning with options attains a better performance than learning with primitive actions, when either the options are carefully designed or the offline data is limited.

**Corollary 1** (Better performance). *Let* $\mathrm{TrueSubOpt}_{\mathcal{D}_1}(\widehat{\mu}, s_1) := V_1^{*,\mathrm{pri}}(s_1) - V_1^{\widehat{\mu}}(s_1)$, *where* $V^{*,\mathrm{pri}}$ *is the optimal value function defined for the primitive actions. Ignoring the concentrability parameters, we have that* $\mathrm{TrueSubOpt}_{\mathcal{D}_1}(\widehat{\mu}, s_1) \leq \tilde{O}(\sqrt{H^5 SC^*/K})$ *attained by the VI-LCB algorithm (Xie et al., 2021), when either the options are carefully designed (i.e.,* $\Delta_{\mathcal{O}}(s_1) := V_1^{*,\mathrm{pri}}(s_1) - V_1^*(s_1) = 0$*) or*

---

[4]We note that the extra factor $H$ in the suboptimality bound (8) can be reduced by applying the reference-advantage decomposition technique (Xie et al., 2021).

*the number $K$ of trajectories in the dataset is*

$$o\left(\frac{H^3}{\Delta_{\mathcal{O}}^2}\left(\sqrt{H^2 S C^*} - \sqrt{C_1^{\text{option}} Z_{\mathcal{O}}^* \overline{Z}_{\mathcal{O}}^*}\right)_+^2\right)$$

*where $(x)_+ := \max\{0, x\}$ for any $x \in \mathbb{R}$.*

Corollary 1 implies that when data is limited, e.g., in cases where the data collection is highly expensive or risky, learning with options is beneficial since the output hierarchical policy yields a higher value than learning with primitive actions.

## 5.2 Learning from State-Action Transitions

We consider dataset $\mathcal{D}_2 := \{(s_h^k, a_h^k, r_h^k)\}_{h\in[H],k\in[K]}$ consisting of state-action-reward tuples, which is collected by an experimenter's interaction with the environment for $K$ episodes using any arbitrary behavior policy. That is, the experimenter takes action $a_h^k$ at state $s_h^k$ at timestep $h$ of the $k$th episode, receives a reward of $r_h^k$, and transits to state $s_{h+1}^k$.

When dataset $\mathcal{D}_2$ is provided, the OOE subroutine in Algorithm 1 is given by Subroutine 3. Note that one difficulty is that we cannot directly estimate the option transition function and the option utility function from dataset $\mathcal{D}_2$ as it only includes the information of the primitive actions. Hence, Subroutine 3 first constructs the empirical transition kernel $\widehat{P} = \{\widehat{P}_h\}_{h\in[H]}$ and the empirical reward function $\widehat{r} = \{\widehat{r}_h\}_{h\in[H]}$ (lines 4-7), and use them to further construct $\widehat{T}_h$ and $\widehat{U}_h$ (lines 8-20). To

---

**Subroutine 3** Offline Option Evaluation (OOE) for Dataset $\mathcal{D}_2$

---

1: **Input:** Dataset $\mathcal{D}_2 = \{(s_h^k, a_h^k, r_h^k)\}_{h\in[H],k\in[K]}$.
2: **Initialize:** $\widehat{U}_{H+1}(\cdot) \leftarrow 0$. Let $N_h(s,a) := \sum_{k=1}^K \mathbb{I}[s_h^k = s, a_h^k = a]$ denote the number of visits of state-action pair $(s,a)$ at timestep $h$ in dataset $\mathcal{D}_2$. Function $\phi := \{\phi_h : \mathcal{S} \times \mathcal{O} \mapsto \mathbb{R}\}_{h\in[H]}$ is given by Equation (32) in the Appendices.
3: **for** $h = H, H-1, ..., 1$ **do**
4:     **for** $(s,a) \in \mathcal{S} \times \mathcal{A}$ **do**
5:         $\widehat{P}_h(s'|s,a) \leftarrow \frac{\sum_{k\in[K]} \mathbb{I}[s_h^k=s, a_h^k=a, s_{h+1}^k=s']}{1 \vee N_h(s,a)}$ for any $s' \in \mathcal{S}$
6:         $\widehat{r}_h(s,a) \leftarrow \mathbb{I}[N_h(s,a) \geq 1] r_h(s,a)$
7:     **end for**
8:     **for** $(s, o, s') \in \mathcal{S} \times \mathcal{O} \times \mathcal{S}$ **do**
9:         $\widehat{T}_h(s'|s,o,1) \leftarrow \beta_{h+1}^o(s') \sum_{a\in\mathcal{A}} \pi_h^o(a|s) \widehat{P}_h(s'|s,a)$
10:         $\overline{\widehat{T}}_h(s'|s,o,1) \leftarrow (1 - \beta_{h+1}^o(s')) \sum_{a\in\mathcal{A}} \pi_h^o(a|s) \widehat{P}_h(s'|s,a)$
11:         **for** $l = 2, \cdots, H-h+1$ **do**
12:             $\widehat{T}_h(s'|s,o,l) \leftarrow \sum_a \pi_h^o(a|s) \sum_{s''} \widehat{P}_h(s''|s,a)\left(1 - \beta_{h+1}^o(s'')\right) \widehat{T}_{h+1}(s'|s'',o,l-1)$
13:             $\overline{\widehat{T}}_h(s'|s,o,l) \leftarrow \sum_a \pi_h^o(a|s) \sum_{s''} \widehat{P}_h(s''|s,a)\left(1 - \beta_{h+1}^o(s'')\right) \overline{\widehat{T}}_{h+1}(s'|s'',o,l-1)$
14:         **end for**
15:     **end for**
16: **end for**
17: **for** $(h, s, o) \in [H] \times \mathcal{S} \times \mathcal{O}$ **do**
18:     $\Gamma_h(s,o) \leftarrow \tilde{O}\left(\sqrt{\sum_{m=h}^H \sum_{(s,a)\in\mathcal{X}_{h,s,o}^m} \frac{H^3 S}{N_m(s,a)\vee 1}} + H\phi_h(s,o)\right)$
19:     $\widehat{U}_h(s,o) \leftarrow \sum_{a\in\mathcal{A}} \pi_h^o(a|s) \widehat{r}_h(s,a) + \sum_{s'\in\mathcal{S}} \overline{\widehat{T}}_h(s'|s,o,1) \widehat{U}_{h+1}(s',o)$
20: **end for**
21: **Output:** $(\widehat{T} = \{\widehat{T}_h\}_{h\in[H]}, \widehat{U} = \{\widehat{U}_h\}_{h\in[H]}, \Gamma = \{\Gamma_h\}_{h\in[H]})$.

---

derive the suboptimality, we first define some useful notations. For any $(h, s, o) \in [H] \times \mathcal{S} \times \mathcal{O}$ and $h \leq m \leq H$, we denote by $\mathcal{X}_{h,s,o}^m$ the set of state-action pairs that can be reached at timestep $m$ by using option $o$ at state $s$ and timestep $h$ without being terminated.[5] Further, let $d^\rho := \{d_h^\rho : \mathcal{S} \times \mathcal{A} \mapsto$

---

[5]For convenience, we assume that $\mathcal{X}_{h,s,o}^m$ is known prior. When $\mathcal{X}_{h,s,o}^m$ is unknown, it can be replaced by a superset that does not require prior knowledge and our results directly follow (See Remark 3 in Appendix F).

$[0,1]\}_{h\in[H]}$ denote the state-action distribution of the behavior policy $\rho$ used by the experimenter. That is, $d_h^\rho(s,a)$ is the probability that the agent takes action $a$ at state $s$ at timestep $h$. Similarly, we make the following assumption on dataset $\mathcal{D}_2$.

**Assumption 2** (Single hierarchical policy concentrability for dataset $\mathcal{D}_2$). The experimenter collects dataset $\mathcal{D}_2$ by following an arbitrary behavior policy $\rho$. There exists some deterministic optimal hierarchical policy $\mu^*$ such that

$$C_2^{\text{option}} := \max_{h,s,o} \sum_{h\leq m\leq H,(s',a')\in\mathcal{X}_{h,s,o}^m} \frac{\theta_h^{\mu^*}(s,o)}{d_m^\rho(s',a')} \tag{9}$$

(with the convention $0/0 = 0$) is finite.

Intuitively, Assumption 2 states that dataset $\mathcal{D}_2$ sufficiently covers the trajectories of state-action-reward tuples induced by the optimal hierarchical policy $\mu^*$. Next, we derive an upper bound of $\text{SubOpt}_{\mathcal{D}_2}$ under Assumption 2. The detailed proof can be found in Appendix F.

**Theorem 5** (Suboptimality for dataset $\mathcal{D}_2$). *Under Assumption 2, with probability at least $1-\xi$, we have that*

$$\text{SubOpt}_{\mathcal{D}_2}(\widehat{\mu},s_1) \leq \tilde{O}\left(\sqrt{\frac{C_2^{\text{option}}H^3SZ_\mathcal{O}^*\overline{Z}_\mathcal{O}^*}{K}} + \frac{H^2SOC_2^{\text{option}}}{K}\right) \tag{10}$$

*which translates to*

$$\tilde{O}\left(\sqrt{\frac{C_2^{\text{option}}H^3SZ_\mathcal{O}^*\overline{Z}_\mathcal{O}^*}{K}}\right)$$

*when $K$ of dataset $\mathcal{D}_2$ is sufficiently large, i.e., $K \geq \tilde{O}(C_2^{\text{option}}H^5S^9A^2O^2/(Z_\mathcal{O}^*\overline{Z}_\mathcal{O}^*))$, where $Z_\mathcal{O}^* = Z_\mathcal{O}^{\mu^*}$ and $\overline{Z}_\mathcal{O}^* = \overline{Z}_\mathcal{O}^{\mu^*}$.*

While, in general, suboptimality bound (10) does not compare favorably against the suboptimality $\tilde{O}(\sqrt{H^5SC^*/K})$ attained by the VI-LCB algorithm that learns with primitive actions, we argue that it can be better in long-horizon problems where the horizon $H$ is much greater than the cardinality of the state space $S$.

## 5.3 Further Discussion

We analyze the pros and cons of both data-collection procedures, which sheds light on offline RL with options in practice. Compared to $\mathcal{D}_2$, dataset $\mathcal{D}_1$ requires smaller storage and enjoys faster convergence to the optimal value, which is further illustrated as follows.

- **Storage:** For dataset $\mathcal{D}_2 = \{(s_h^k, a_h^k, r_h^k)\}_{h\in[H],k\in[K]}$, the storage is simply $HK$. However, for dataset $\mathcal{D}_1 = \{(s_{t_i^k}^k, o_{t_i^k}^k, u_{t_i^k}^k)\}_{i\in[j^k],k\in[K]}$, its expected size is $K \cdot Z_\mathcal{O}^\rho \leq HK$, where $\rho$ is the hierarchical behavior policy. Therefore, in the case of a small $Z_\mathcal{O}^\rho$, dataset $\mathcal{D}_1$ requires much smaller storage than $\mathcal{D}_2$ (on average).

- **Suboptimality:** Ignoring the concentrability, the suboptimality bound (10) for dataset $\mathcal{D}_2$ is worse than the suboptimality bound (8) for dataset $\mathcal{D}_1$ by a factor of $\sqrt{S}$, which is introduced when estimating the option transition function and the option utility function using only the information of the primitive actions.

However, since dataset $\mathcal{D}_2$ contains more information on the environment than $\mathcal{D}_1$, it has a weaker requirement on the behavior (hierarchical) policy and allows the evaluation of new options, which is illustrated as follows.

- **Concentrability:** Recall that the suboptimality bounds for both datasets build upon the sufficient coverage assumptions, i.e., Assumption 1 for $\mathcal{D}_1$ and Assumption 2 for $\mathcal{D}_2$. While they are generally incomparable (as dataset $\mathcal{D}_2$ can be collected by an arbitrary behavior policy), we focus on the case that both datasets are collected by the same hierarchical behavior policy $\rho$. Particularly, it can be shown that Assumption 2 is weaker than Assumption 1. Indeed, if $\rho$

covers the trajectories of state-option-utility tuples induced by $\mu^*$ (i.e., Assumption 1 holds), then it must have covered the trajectories of state-action-reward tuples induced by $\mu^*$ (i.e., Assumption 2 holds). However, the opposite does not hold in general and we provide such an example in Appendix H.

- **Evaluation of New Options:** In the options literature, a popular task is offline option discovery (Ajay et al., 2021; Villecroze et al., 2022), i.e., designing new and useful options from the dataset. Therefore, an important problem is whether these new options can be evaluated through the dataset. We argue that dataset $\mathcal{D}_2$ yields greater flexibility than $\mathcal{D}_1$ in this case. Again, we assume that both datasets are collected by the same hierarchical behavior policy $\rho$. Unfortunately, one cannot use dataset $\mathcal{D}_1$ to evaluate any $(h, s, o)$ that is not visited by $\rho$, i.e., $\theta_h^\rho(s, o) = 0$, let along evaluating the new options. However, this is not the case for dataset $\mathcal{D}_2$. In fact, any $(h, s, o)$ can be evaluated if the visiting state-action pairs are also reachable by $\rho$, i.e., $\sum_{h \leq m \leq H, (s', a') \in \mathcal{S}_{h,s,o}^m} 1/d_m^\rho(s', a') < \infty$. An interesting problem is how to leverage the results in this paper to facilitate offline option discovery, which we shall research in the future work.

## 6 Conclusions

In this paper, we provide the first analysis of the sample complexity for offline RL with options. A novel information-theoretic lower bound is established, which generalizes the one for offline RL with actions. We derive near-optimal suboptimality bounds of the **PE**ssimistic **V**alue **I**teration for Learning with **O**ptions (PEVIO) algorithm for two popular data-collection procedures. Our results show that options facilitate more sample-efficient learning than primitive actions in offline RL in both the finite-time convergence rate to the optimal value and the actual performance.

## Acknowledgments and Disclosure of Funding

The work presented in this paper was partially supported by a research grant from the Research Grants Council, Hong Kong, China (RGC Ref. No. CUHK 14206820). Part of the research reported in this paper was done when the second author was with The Chinese University of Hong Kong. The authors are grateful to the anonymous reviewers for their helpful comments.

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
