**Lemma 1.** *Consider $T_h(s'|s, o, \tau)$ defined in Equation (11), it holds that*

$$\sum_{\tau \in [H-h+1]} \sum_{s' \in \mathcal{S}} T_h(s'|s, o, \tau) = 1 \tag{12}$$

*for any $(h, s) \in [H] \times \mathcal{S}$.*

Recall that for any $(s', l) \in \mathcal{S} \times [H - h + 1]$, $\overline{T}_h(s'|s, o, l)$ is the probability that the agent uses option $o$ at state $s$ at timestep $h$, reaches state $s'$ at timestep $h + l$ without being terminated option $o$ in these $l$ timesteps, but does *not* terminate option $o$ at state $s'$ at timestep $h + l$. Hence, we can recursively define it by

$$\overline{T}_h(s'|s, o, l) := \sum_{a \in \mathcal{A}} \pi_h^o(a|s) \sum_{s'' \in \mathcal{S}} P_h(s''|s, a) \left(1 - \beta_{h+1}^o(s'')\right) \overline{T}_{h+1}(s'|s'', o, l - 1) \tag{13}$$

where $\overline{T}_h(s'|s, o, 1) := (1 - \beta_{h+1}^o(s')) \sum_{a \in \mathcal{A}} \pi_h(a|s) P_h(s'|s, a)$. Intuitively, $T_h(s'|s, o, \tau)$ can be interpreted as the joint probability of two consecutive and independent events: (i) at state $s$ at timestep $h$, the agent uses option $o$ for $t < \tau$ timesteps without being terminated this option, reaches some state $s_{h+t}$, and does not terminate option $o$ at state $s_{h+t}$ at timestep $h + t$ (the probability of this event is $\overline{T}_h(s_{h+t}|s, o, t)$), (ii) at state $s_{h+t}$ at timestep $h + t$, the agent keeps using option $o$ for $(\tau - t)$ timesteps, reaches state $s'$, and then terminates option $o$ at state $s'$ at timestep $h + \tau$ (the probability of this event is $T_{h+t}(s'|s_{h+t}, o, \tau - t)$). Since events (i) and (ii) are independent, the joint probability is therefore $\overline{T}_h(s_{h+t}|s, o, t) T_{h+t}(s'|s_{h+t}, o, \tau - t)$. Similarly, $\overline{T}_h(s'|s, o, \tau)$ can be thought of as the joint probability of two consecutive and independent events: (i') at state $s$ at timestep $h$, the agent uses option $o$ for $t < \tau$ timesteps without being terminated this option, reaches some state $s_{h+t}$, and does not terminate option $o$ at state $s_{h+t}$ at timestep $h + t$ (the probability of this event is $\overline{T}_h(s_{h+t}|s, o, t)$), (ii') at state $s_{h+t}$ at timestep $h + t$, the agent keeps using option $o$ for $(\tau - t)$ timesteps, reaches state $s'$, and does not terminates option $o$ at state $s'$ at timestep $h + \tau$ (the probability of this event is $\overline{T}_{h+t}(s'|s_{h+t}, o, \tau - t)$). Again, since events (i') and (ii') are independent, the joint probability is $\overline{T}_h(s_{h+t}|s, o, t) \overline{T}_{h+t}(s'|s_{h+t}, o, \tau - t)$. We formalize this idea in the following lemma. (The detailed proof can be found in Appendix G.2.)

**Lemma 2.** *Consider $T_h(s'|s, o, \tau)$ and $\overline{T}_h(s'|s, o, l)$ defined in Equation (11) and Equation (13), respectively, it holds that*

$$T_h(s'|s, o, 1) + \overline{T}_h(s'|s, o, 1) = \sum_{a \in \mathcal{A}} \pi_h^o(a|s) P_h(s'|s, a) \tag{14}$$

$$T_h(s'|s, o, \tau) = \sum_{s'' \in \mathcal{S}} \overline{T}_h(s''|s, o, t) T_{h+t}(s'|s'', o, \tau - t) \tag{15}$$

$$\overline{T}_h(s'|s, o, l) = \sum_{s'' \in \mathcal{S}} \overline{T}_h(s''|s, o, t) \overline{T}_{h+t}(s'|s'', o, l - t) \tag{16}$$

*for any $\tau \geq 2$ and $t \in [\tau - 1]$.*

In addition, recall that $U_h(s, o) = \mathbb{E}[u_h(s, o)]$, where $u_h(s, o)$ is the random cumulative reward within timesteps that the agent keeps using option $o$ from timestep $h$ without being terminated it, provided that the agent is at state $s$ at timestep $h$. Hence, it can be recursively defined as

$$
\begin{aligned}
U_h(s, o) &= \sum_{a \in \mathcal{A}} \pi_h^o(a|s) r_h(s, a) + \sum_{s' \in \mathcal{S}} \overline{T}_h(s'|s, o, 1) U_{h+1}(s', o) \\
&= \sum_{a \in \mathcal{A}} \pi_h^o(a|s) r_h(s, a) + \sum_{s' \in \mathcal{S}} \sum_{a' \in \mathcal{A}} \overline{T}_h(s'|s, o, 1) \pi_{h+1}^o(a'|s') r_{h+1}(s', a') \\
&\quad + \sum_{s'' \in \mathcal{S}} \overline{T}_h(s''|s, o, 2) U_{h+2}(s'', o) \\
&= \cdots = \sum_{a \in \mathcal{A}} \pi_h^o(a|s) r_h(s, a) + \sum_{l \in [H-h]} \sum_{s' \in \mathcal{S}} \sum_{a' \in \mathcal{A}} \overline{T}_h(s'|s, o, l) \pi_{h+l}^o(a'|s') r_{h+l}(s', a')
\end{aligned}
\tag{17}
$$

where $U_H(s, o) := \sum_{a \in \mathcal{A}} \pi_H^o(a|s) r_H(s, a)$. Finally, for any hierarchical policy $\mu$ and an episode starts from state $s_1$ at the first timestep, we define the state-option occupancy measure $\theta^\mu$ as follows.

$$
\theta_h^\mu(s, o) := \mu_h(o|s) \sum_{\tau \in [h-1]} \sum_{s' \in \mathcal{S}} \sum_{o' \in \mathcal{O}} \theta_{h-\tau}^\mu(s', o') T_{h-\tau}(s|s', o', \tau)
\tag{18}
$$

where $\theta_1(s, o) := \mathbb{I}[s = s_1] \mu_1(o|s)$.

## A.2    Notations in Section 5.2

Recall that $d_h^\rho(s, a)$ is the probability that the agent takes action $a$ at state $s$ at timestep $h$, given that the episode starts from state $s_1$ at the first timestep. When the behavior policy $\rho = \{\rho_h : \mathcal{S} \mapsto \Delta(\mathcal{A})\}_{h \in [H]}$ is not hierarchical, $d_h^\rho(s, a)$ is given as

$$
d_h^\rho(s, a) := \rho_h(a|s) \sum_{s' \in \mathcal{S}} \sum_{a' \in \mathcal{A}} P_{h-1}(s|s', a') d_{h-1}^\rho(s', a')
\tag{19}
$$

where $d_1^\rho(s, a) = \mathbb{I}[s = s_1] \rho_1(a|s)$. When the behavior policy $\rho = \{\rho_h : \mathcal{S} \mapsto \Delta(\mathcal{O})\}_{h \in [H]}$ is hierarchical, we first define the probability $q_h^\mu(s, o)$ that the agent use option $o$ at state $s$ at timestep $h$ following the hierarchical policy $\mu$, given that the episode starts at state $s_1$ at the first timestep. That is,

$$
q_h^\mu(s, o) := \sum_{s' \in \mathcal{S}} \sum_{o' \in \mathcal{O}} q_{h-1}^\mu(s', o') \sum_{a' \in \mathcal{A}} \pi_{h-1}^{o'}(a'|s') P_{h-1}(s|s', a') \left( \mathbb{I}[o' = o](1 - \beta_h^{o'}(s)) + \beta_h^{o'}(s) \mu_h(o|s) \right)
$$

where $q_1^\mu(s, o) = \mathbb{I}[s = s_1] \mu_1(o|s)$. We note that the difference between $q_h^\mu(s, o)$ and $\theta_h^\mu(s, o)$ defined in Equation (18) is that $q_h^\mu(s, o)$ does not require option $o$ to be newly selected at timestep $h$, i.e., option $o_{h-1}$ needs not to be terminated. Therefore, $d_h^\rho(s, a)$ for a hierarchical policy $\rho$ is given by

$$
d_h^\rho(s, a) := \sum_{o \in \mathcal{O}} q_h^\mu(s, o) \mu_h(a|s)
$$

## B    Proof of Theorem 1

*Proof.* We decompose the $Q$-function for any $(h, s, o) \in [H] \times \mathcal{S} \times \mathcal{O}$ as follows.

$$
\begin{aligned}
Q_h^\mu(s, o) &= \mathbb{E}_\mu \left[ \sum_{h'=h}^H r_{h'}(s_{h'}, a_{h'}) \middle| s_h = s, o_h = o \right] \\
&= \underbrace{\sum_{a \in \mathcal{A}} \pi_h^o(a|s) r_h(s, a)}_{(7.1)} + \underbrace{\sum_{s' \in \mathcal{S}} \left( T_h(s'|s, o, 1) V_{h+1}^\mu(s') + \overline{T}_h(s'|s, o, 1) Q_{h+1}^\mu(s', o) \right)}_{(7.2)}
\end{aligned}
\tag{20}
$$

and we define $V_{H+1}^\mu(s) = Q_{H+1}^\mu(s, o) = 0$ for any $(s, o) \in \mathcal{S} \times \mathcal{O}$. That is to say, the $Q$-function (20) is the expected return from timestep $h$ to timestep $H$, provided that option $o$ is used at state $s$ at

timestep $h$, no matter option $o_{h-1}$ used at timestep $h-1$ is terminated or not. Similar to classic RL, the $Q$-function for learning with options can be decomposed into two parts. Term (7.1) is the (expected) instant reward at timestep $h$ and term (7.2) is the expected return in the rest of the episode (from timestep $h+1$ to timestep $H$). However, different from the $Q$-function for classic RL with primitive actions, the expected return in the rest of the episode is not the expectation of the value function over the states at timestep $h+1$. Intuitively, after some state $s'$ is reached at timestep $h+1$ with probability $\sum_{a \in \mathcal{A}} \pi_h^o(a|s) P_h(s'|s,a)$ given that option $o$ is used at state $s$ at timestep $h$, only one of the following two situations happens. The first situation is that option $o$ is not terminated at state $s'$ at timestep $h+1$ (the probability of this event is $\overline{T}_h(s'|s,o,1)$). As a result, the agent uses option $o$ at state $s'$ at timestep $h+1$, and hence the expected return in the rest of the episode conditioned on this first situation is $Q_{h+1}^\mu(s',o)$. The second situation is that option $o$ is terminated at state $s'$ at timestep $h+1$ (the probability of this event is $T_h(s'|s,o,1)$, and by Equation (14) we have that $T_h(s'|s,o,1) = \sum_{a \in \mathcal{A}} \pi_h^o(a|s) P_h(s'|s,a) - \overline{T}_h(s'|s,o,1)$). Consequently, the agent selects a new option according to the hierarchical policy at state $s'$ at timestep $h+1$, and hence the expected return in the rest of the episode conditioned on this second situation is $V_{h+1}^\mu(s')$. Note that when $\mathcal{O} = \mathcal{A}$ (In this case, we have that $\overline{T}_h(s'|s,o,1) = 0$ hold for any $(h,s,o) \in [H] \times \mathcal{S} \times \mathcal{O}$),[6] Equation (20) recovers the $Q$-function for episodic MDP that learns with primitive actions, e.g. (Jin et al., 2021, Equation (2.5)). Further, by iteratively applying Equation (20) over $h+1, \cdots, H$, we derive that

$$
\begin{aligned}
Q_h^\mu(s,o) = &\sum_{a \in \mathcal{A}} \pi_h^o(a|s) \left( r_h(s,a) + \sum_{s' \in \mathcal{S}} \overline{T}_h(s'|s,o,1) \sum_{a' \in \mathcal{A}} \pi_{h+1}^o(a'|s') r_{h+1}(s',a') \right) \\
&+ \sum_{s'' \in \mathcal{S}} \underbrace{\sum_{s' \in \mathcal{S}} \overline{T}_h(s'|s,o,1) T_{h+1}(s''|s',o,1)}_{= T_h(s''|s,o,2) \text{ by Equation (15)}} V_{h+2}^\mu(s'') \\
&+ \sum_{s'' \in \mathcal{S}} \underbrace{\sum_{s' \in \mathcal{S}} \overline{T}_h(s'|s,o,1) \overline{T}_{h+1}(s''|s',o,1)}_{= \overline{T}_h(s''|s,o,2) \text{ by Equation (16)}} Q_{h+2}^\mu(s'',o) \\
&+ \sum_{s' \in \mathcal{S}} T_h(s'|s,o,1) V_{h+1}^\mu(s') \\
= &\cdots = U_h(s,o) + \sum_{\tau \in [H-h+1]} [T_h V_{h+\tau}^\mu](s,o)
\end{aligned}
\tag{21}
$$

where $[T_h y_{h+\tau}](s,o) := \sum_{s' \in \mathcal{S}} T_h(s'|s,o,\tau) y_{h+\tau}(s')$ is defined for any $(s,o,\tau) \in \mathcal{S} \times \mathcal{O} \times [H - h + 1]$ given any arbitrary series of functions $\{y_h : \mathcal{S} \mapsto \mathbb{R}\}_{h \in [H]}$. The value function for any $(h,s) \in [H] \times \mathcal{S}$ can be derived as

$$
V_h^\mu(s) := \mathbb{E}_\mu \left[ \sum_{h'=h}^H r_{h'}(s_{h'}, a_{h'}) \middle| s_h = s, o_h \sim \mu_h(\cdot|s_h) \right] = \sum_{o \in \mathcal{O}} \mu_h(o|s) Q_h^\mu(s,o)
$$

which concludes the proof. $\qquad\square$

## C  Proof of Theorem 2

*Proof.* Our construction of the hard episodic MDP instance is inspired by Xie et al. (2021, Appendix D).

**Construction of hard instances.** The family of MDPs has $(S+2)$ states, $(2H+1)$ timesteps, $A$ actions, and $(O+1)$ options for any $S, H, O \geq 1$ (The rescaling only affects $H, S$ by at most a multiplicative constant and thus does not affect the result). Each MDP $M_{\mathbf{a}^*}(\mathcal{S}, \mathcal{A}, \mathcal{O}, H, \mathcal{P}, r)$ is index by a vector $\mathbf{a}^* = (a_{h,i}^*) \in [A]^{HS}$ and is specified as follows.

---

[6]The notation $\mathcal{O} = \mathcal{A}$ means that for any $a \in \mathcal{A}$, there exists $o \in \mathcal{O}$ such that $\pi_h^o(a|s) = 1$ and $\beta_h^o(s) = 1$ for any $(h,s,o) \in [H] \times \mathcal{S} \times \mathcal{O}$. In addition, it holds that $O = |\mathcal{O}| = |\mathcal{A}| = A$.

- The state space is $\mathcal{S} := \{s_i\}_{i\in[S]} \cup \{s_g, s_b\}$. There are $S$ "bandit states" $\{s_i\}_{i\in[S]}$, one "good state" $s_g$, and one "bad state" $s_b$. Particularly, $s_g$ and $s_b$ are absorbing states, i.e., $P_h(s_g|s_g, a) = P_h(s_b|s_b, a) = 1$ for all $h \in [2H+1]$ and all $a \in [A]$.

- The action space is $\mathcal{A} := [A]$.

- The option set is $\mathcal{O} := \{o_j\}_{j\in[O]} \cup \{o^*\}$.

  - The option $o^*$ satisfies that $\pi_h^{o^*}(a_{h,i}^*|s_i) = 1$ for any $(h,i) \in [H+1] \times [S]$ and $\pi_h^{o^*}(1|s) = 1$ whenever $h \geq H+1$ or $s \in [s_g, s_b]$, i.e., option $o^*$ always takes action $a_{h,i}^*$ at each bandit state $s_i$ and otherwise, takes action 1.

  - For any $o \neq o^*$, it holds that $\pi_h^o(j|s_i) = 1$ for some $j \neq a_{h,i}^*$ and any $(h,i) \in [\lfloor z^* \rfloor] \times [S]$, i.e., any option other than $o^*$ always takes the suboptimal action at each bandit state at the first $\lfloor z^* \rfloor$ timesteps. And at timestep $\lfloor z^* \rfloor + 1 \leq h \leq H+1$, we have that $\pi_h^o(a_{h,i}^*|s_i) = 1$ for any $o \neq o^*$. In addition, it holds that $\pi_h^o(1|s) = 1$ whenever $h \geq H+1$ or $s \in [s_g, s_b]$.

  - Further, for any $o \in \mathcal{O}$, it holds that $\beta_h^o(s_i) = 1$ for any $(h,i) \in [\lfloor z^* \rfloor] \times [S]$ and $\beta_{h'}^o(s_i) = 0$ for any $([2H+1]\backslash[\lfloor z^* \rfloor]) \times [S]$, i.e., any option is guaranteed to terminate (continue) at any bandit state and at each timestep $h \in [\lfloor z^* \rfloor]$ ($h' \in ([2H+1]\backslash[\lfloor z^* \rfloor])$). In addition, each option is continued at state $s_g$ or $s_b$, i.e., $\beta_h^o(s_g) = \beta_h^o(s_b) = 0$ for any $(h, o) \in [2H+1] \times \mathcal{O}$.

- Transition kernel $\mathcal{P}$: At the first $H$ timesteps, each bandit state $s_i$ can only transit to $s_i$ itself, $s_g$, or $s_b$. The transition probabilities satisfy: (i) $P_h(s_i|s_i, a) = 1 - \frac{1}{H}$ for all $a \in [A]$, (ii) $P_h(s_g|s_i, a) = P_h(s_b|s_i, a) = \frac{1}{2H}$ for any $a \neq a_{h,i}^*$, and (iii) $P_h(s_g|s_i, a_{h,i}^*) = \frac{1}{H}(\frac{1}{2} + \tau), P_h(s_b|s_i, a_{h,i}^*) = \frac{1}{H}(\frac{1}{2} - \tau)$, where $\tau$ is a parameter to be determined. At the last $H+1$ timesteps, all bandit states transit to one of $s_g$ and $s_b$ with probability $1/2$ each.

- Reward function $r$: The bandit states and the bad state $s_g$ do not receive any reward, i.e., $r_h(s_i, a) = r_h(s_g, a) = 0$ for any $(h,i,a) \in [2H+1] \times [S] \times [A]$. The good state does not receive any reward at the first $H+1$ timesteps, while for $h \geq H+2$, it receives a reward 1 regardless of the action taken, i.e., $r_h(s_g, a) = 0$ for any $(h,a) \in [H+1] \times [A]$ and $r_h(s_g, a) = 1$ for any $(h,a) \in ([2H+1]\backslash[H+1]) \times [A]$.

- Initial state distribution is uniform on all bandit states, i.e., $S_1 \sim \text{Unif}\{s_i\}_{i\in[S]}$.

We also let $M_0$ denote the "null" MDP that has the same construction as the above except that there is no "special" action $a_{h,i}^*$, that is, for any $a \in [A]$, it holds that

$$P_h(s_g|s_i, a) = P_h(s_b|s_i, a) = \frac{1}{2H}$$

**Optimal hierarchical policy $\mu^*$ and hierarchical behavior policy $\rho$.** We define our deterministic hierarchical policy $\mu^* = \{\mu_h^*(s) = o^* \text{ for any } s \in \mathcal{S}\}_{h\in[H]}$. Therefore, we have that $\sum_{h,s,o} \theta_h^{\mu^*}(s, o) \leq \lfloor z^* \rfloor \leq z^*$ and $\sum_{h,s,o} \mathbb{I}[\theta_h^{\mu^*}(s, o) > 0] \leq S\lfloor z^* \rfloor \leq \overline{z}^*$. Let $B = \lfloor C^{\text{option}} \rfloor$ denote the largest integer no greater than $C^{\text{option}}$. The hierarchical behavior policy $\rho$ satisfies that $\rho_h(o|s_i) = \frac{1}{B}$ for any $o \in \{o^*\} \cup \{o_j\}_{j\in[B-1]}$ and all $(h,i) \in [H] \times [S]$ and $\rho_h(o^*|s) = 1$ whenever $h \geq H+1$ or $s \in \{s_g, s_b\}$. It should be obvious that

$$\max_{h,s,o} \frac{\theta_h^{\mu^*}(s, o)}{\theta_h^{\rho}(s, o)} \leq B \leq C^{\text{option}}$$

Since the above statement holds for any arbitrary selection of $a^* \in [B]^{HS}$. Therefore, the following family of problems is indeed a subset of the class $\mathcal{M}(C^{\text{option}}, z^*, \overline{z}^*)$.

$$\{(M_{a^*}, \rho) : a^* \in [K]^{HS}\} \subset \mathcal{M}(C^{\text{option}}, z^*, \overline{z}^*)$$

We denote by $\nu$ the uniform (prior) distribution on $[B]^{HS}$, i.e., $\nu(a^* = a_0) = 1/B^{HS}$ for all $a_0 \in [B]^{HS}$. Note that the hierarchical behavior policy $\rho$ is the same for all MDPs in the above

family of problems. Define

$$l_{h,i}(\hat{\mu}, \mathbf{a}^*) := \mathbb{P}_{\mathbf{a}^*}(\hat{\mu}_h(s_i) \neq o^*), \quad L(\hat{\mu}, \mathbf{a}^*) := \sum_{h=1}^{\lfloor z^* \rfloor} \sum_{i=1}^{S} l_{h,i}(\hat{\mu}, \mathbf{a}^*)$$

The loss $L$ measures the expected number of $(h, i)$ pairs on which the algorithm fails to identify the best option $o^*$. A large loss will translate to a high suboptimality bound, which is formalized in the following lemma (The detailed proof can be found in Appendix G.3).

**Lemma 3.** *For any $\mathbf{a}^* \in [B]^{HS}$ and any offline algorithm that outputs a deterministic hierarchical policy $\hat{\mu}$, we have that*

$$\mathbb{E}_{M_{\mathbf{a}^*}} \left[ V_{1,M_{\mathbf{a}^*}}^* - V_{1,M_{\mathbf{a}^*}}^{\hat{\mu}} \right] \geq \frac{\tau}{3S} \cdot L(\hat{\mu}, \mathbf{a}^*) \tag{22}$$

Next, we devote to establishing the following inequality, which lower bounds $L(\hat{\mu}, \mathbf{a}^*)$.

$$\mathbb{E}_{\mathbf{a}^* \sim \nu}[L(\hat{\mu}, \mathbf{a}^*)] \geq \lfloor z^* \rfloor S \left( \frac{1}{2} - \sqrt{\frac{2\tau^2 K}{BHS}} \right) \tag{23}$$

Once Inequality (23) is established, then if $\sqrt{2\tau^2 K / BHS} \leq \frac{1}{4}$, we have that

$$\mathbb{E}_{\mathbf{a}^* \sim \nu}[L(\hat{\mu}, \mathbf{a}^*)] \geq \frac{\lfloor z^* \rfloor S}{4}$$

Plugging in Inequality (22), we can further derive that

$$\mathbb{E}_{\mathbf{a}^* \sim \nu}[L(\hat{\mu}, \mathbf{a}^*)] \geq \frac{\tau}{3S} \frac{\lfloor z^* \rfloor S}{4} = \frac{\tau \lfloor z^* \rfloor}{12}$$

Let $\tau = 12\epsilon / \lfloor z^* \rfloor$ where $\epsilon \leq 1/12$. Then, if

$$K \leq \frac{BHS}{32\tau^2} = \frac{BHS(\lfloor z^* \rfloor)^2}{32 \cdot 12^2 \epsilon^2} = c_0 \cdot \frac{C^{\text{option}} H z^* \overline{z}^*}{\epsilon^2}$$

Hence, the suboptimality satisfies that

$$\mathbb{E}_{\mathbf{a}^* \sim \nu}[L(\hat{\mu}, \mathbf{a}^*)] \geq \epsilon$$

and we conclude the proof of Theorem 2. Therefore, the rest of this section is to establish Inequality (23). By slightly modifying the proof of (Xie et al., 2021, Theorem 3), we have that

$$\mathbb{E}_{\mathbf{a}^* \sim \nu}[l_{h,i}(\hat{\mu}, \mathbf{a}^*)] \geq \frac{1}{2} - \sqrt{\mathbb{E}_{\mathbb{P}_o, \mathcal{A}}[N_h(s_i)] \cdot 2\tau^2 / HB}$$

Then, summing the preceding bound over all $(h, i)$, we have that for the hierarchical policy $\hat{\mu}$ output by any offline algorithm,

$$\mathbb{E}_{\mathbf{a}^* \sim \nu}[L(\hat{\mu}, \mathbf{a}^*)]$$

$$\geq \lfloor z^* \rfloor S \left( \frac{1}{2} - \sqrt{\frac{1}{\lfloor z^* \rfloor S} \sum_{h=1}^{\lfloor z^* \rfloor} \sum_{i=1}^{S} \mathbb{E}_{\mathbb{P}_o, \mathcal{A}}[N_h(s_i)] \cdot 2\tau^2 / HB} \right) = \lfloor z^* \rfloor S \left( \frac{1}{2} - \sqrt{\frac{2\tau^2 K}{BHS}} \right)$$

which concludes the proof. $\square$

## D   Proof of Theorem 3

*Proof.* Our proof relies on the following lemma, which decomposes the suboptimality (3) into three terms that can be analyzed separately. (The detailed proof can be found in Appendix G.4.)

**Lemma 4** (Decomposition of suboptimality). *Let $\widehat{\mu} = \{\widehat{\mu}_h\}_{h\in[H]}$, $\widehat{Q} = \{\widehat{Q}_h\}_{h\in[H]}$, $\widehat{V} = \{\widehat{V}_h\}_{h\in[H]}$ denote the hierarchical policy, the estimated Q-function, and the corresponding estimated value function output from Algorithm 1, respectively. We have that*

$$
\text{SubOpt}_{\mathcal{D}}(\widehat{\mu}, s_1)
$$

$$
= \underbrace{- \sum_{h=1}^{H} \mathbb{E}_{\widehat{\mu}}[\iota_h(s_h, o_h)|s_1]}_{(i)} + \underbrace{\sum_{h=1}^{H} \mathbb{E}_{\mu^*}[\iota_h(s_h, o_h)|s_1]}_{(ii)}
$$

$$
+ \underbrace{\sum_{h=1}^{H} \mathbb{E}_{\mu^*}[\langle \widehat{Q}_h(s_h, \cdot), \mu_h^*(\cdot|s_h) - \widehat{\mu}_h(\cdot|s_h)\rangle_{\mathcal{O}}|s_1]}_{(iii)} \tag{24}
$$

*where $\mu^*$ is the optimal hierarchical policy and*

$$
\iota_h(s, o) = U_h(s, o) + \sum_{\tau \in [H-h+1]} [T_h \widehat{V}_{h+\tau}](s, o) - \widehat{Q}_h(s, o) \tag{25}
$$

*is defined for any $(h, s, o) \in [H] \times \mathcal{S} \times \mathcal{O}$.*

The following lemma states that $\iota$ is non-negative. (The detailed proof can be found in Appendix G.6.)

**Lemma 5** (Pessimism for Learning with Options in General MDP Using Dataset $\mathcal{D}$). *Suppose that $\{\Gamma_h\}_{h\in[H]}$ in Algorithm 1 are $\xi$-uncertainty quantifiers. Under $\mathcal{E}$ defined in Equation (5), which satisfies $\mathbb{P}_{\mathcal{D}}(\mathcal{E}) \geq 1 - \xi$, we have*

$$
0 \leq \iota_h(s, o) \leq 2\Gamma_h(s, o)
$$

*for all $(h, s, o) \in [H] \times \mathcal{S} \times \mathcal{O}$.*

Therefore, the first term Equation (24) non-positive. By Lemma 4 and the fact that $\widehat{\mu}$ is greedy with respect to $\widehat{Q}_h$ (which implies the third term in Equation (24) is non-positive) for all $h \in [H]$, we have that

$$
\text{SubOpt}_{\mathcal{D}}(\widehat{\mu}, s_1) \leq 2 \sum_{h=1}^{H} \mathbb{E}_{\mu^*}[\Gamma_h(s_h, o_h)|s_1]
$$

which concludes the proof. $\qquad\square$

# E  Proof of Theorem 4

*Proof.* We first show that $\Gamma$ defined in line 5 of Subroutine 2 is a $\xi$-uncertainty quantifier. (The detailed proof can be found in Appendix G.7.)

**Lemma 6.** *Given a dataset $\mathcal{D}_1$, we have that*

$$
\Gamma_h = O\left(\sqrt{\frac{H^2}{n_h(s, o) \vee 1} \log\left(\frac{HSO}{\xi}\right)}\right) \tag{26}
$$

*for any $(h, s, o) \in [H] \times \mathcal{S} \times \mathcal{O}$ is a $\xi$-uncertainty quantifier.*

Hence, by Theorem 3 and Lemma 6, we have that with probability at least $1 - \xi$

$$
\text{SubOpt}_{\mathcal{D}_1}(\widehat{\mu}, s_1) \leq \tilde{O}\left(\sum_{h=1}^{H} \mathbb{E}_{\mu^*}\left[\sqrt{\frac{H^2}{n_h(s, o) \vee 1}} \bigg| s_1\right]\right)
$$

$$
\leq \tilde{O}\left(\sum_{h=1}^{H} \mathbb{E}_{\mu^*}\left[\sqrt{\frac{H^3}{K\theta_h^\rho(s, o)}} \bigg| s_1\right]\right) \tag{27}
$$

$$\leq \tilde{O}\left(\sum_{h=1}^{H}\sum_{(s,o)\in\mathcal{S}\times\mathcal{O}}\theta_h^{\mu^*}(s,o)\sqrt{\frac{H^3}{K\theta_h^\rho(s,o)}}\right)$$

$$\leq \tilde{O}\left(\sum_{h=1}^{H}\sum_{(s,o)\in\mathcal{S}\times\mathcal{O}}\sqrt{\frac{\theta_h^{\mu^*}(s,o)C_1^{\text{option}}H^3}{K}}\right) \tag{28}$$

$$\leq \sqrt{\frac{C_1^{\text{option}}H^3}{K}}\cdot\tilde{O}\left(\sqrt{\sum_{h=1}^{H}\sum_{s,o}\mathbb{I}[\theta_h^{\mu^*}(s,o)>0]}\cdot\sqrt{\sum_{h=1}^{H}\sum_{s,o}\theta_h^{\mu^*}(s,o)}\right) \tag{29}$$

$$= \tilde{O}\left(\sqrt{\frac{C_1^{\text{option}}H^3\overline{Z}_{\mathcal{O}}^*Z_{\mathcal{O}}^*}{K}}\right)$$

where $Z_{\mathcal{O}}^* = \sum_{h\in[H]}\sum_{s\in\mathcal{S}}\sum_{o\in\mathcal{O}}\theta_h^{\mu^*}(s,o)$ and $\overline{Z}_{\mathcal{O}}^* = \sum_{h\in[H]}\sum_{s\in\mathcal{S}}\sum_{o\in\mathcal{O}}\mathbb{I}[\theta_h^{\mu^*}(s,o)>0]$. Here, Inequality (27) follows from (Xie et al., 2021, Lemma B.1) with $n=K, p=\theta_h^\rho(s,o)$. Inequality (28) holds by Assumption 1. Inequality (29) is implied by Cauchy-Schwarz Inequality. Therefore, we conclude the proof. $\square$

# F  Proof of Theorem 5

*Proof.* To begin with, we first define the set $\mathcal{X}_{h,s,o}^m$, which is the set of state-action pairs that can be reached at timestep $m$ by using option $o$ at state $s$ and timestep $h$ without being terminated. For $m>h$, $\mathcal{X}_{h,s,o}^m$ is given by

$$\mathcal{X}_{h,s,o}^m := \{(s',a'):\overline{T}_h(s'|s,o,m-h)\cdot\pi_m^o(a'|s')>0\} \tag{30}$$

where $\overline{T}_h(s'|s,o,l)$ is defined in Equation (13). Particularly, we define $\mathcal{X}_{h,s,o}^h := \{(s,a'):\pi_h^o(a'|s)>0\}$.

*Remark* 3. In the proof, we assume that $\mathcal{X}_{h,s,o}^m$ is known prior for convenience. However, this assumption can be relaxed since one can replace $\mathcal{X}_{h,s,o}^m$ with its superset $\overline{\mathcal{X}}_{h,s,o}^m := \{(s',a'):\pi_m^o(a'|s')>0\}$, which does not require prior knowledge, and our results follow directly.

First, we show that $\Gamma$ defined in Line 18 of Subroutine 3 is a $\xi$-uncertainty quantifier. (The detailed proof can be found in Appendix G.8.)

**Lemma 7.** *Given a dataset $\mathcal{D}_2$, we have that*

$$\Gamma_h(s,o) = \tilde{O}\left(H\sqrt{\sum_{m=h}^{H}\sum_{(s,a)\in\mathcal{X}_{h,s,o}^m}\frac{HS}{N_m(s,a)\vee 1}}+H\phi_h(s,o)\right) \tag{31}$$

*where*

$$\phi_h(s,o) := HS^2\sum_{h\leq m<t\leq H}\sqrt{\sum_{(s_m,a_m)\in\mathcal{X}_{h,s,o}^m}\frac{1}{N_m(s_m,a_m)\vee 1}}\cdot\sqrt{\sum_{(s_t',a_t')\in\mathcal{X}_{h,s,o}^t}\frac{1}{N_t(s_t',a_t')\vee 1}}$$

$$+ HS^3\sum_{h\leq m<t\leq H}\sum_{(s_m,a_m)\in\mathcal{X}_{h,s,o}^m,(s_t',a_t')\in\mathcal{X}_{h,s,o}^t}\left(\frac{1}{N_m(s_m,a_m)\vee 1}+\frac{1}{N_t(s_t',a_t')\vee 1}\right)$$

$$+ \sum_{m=h}^{H}\sum_{(s,a)\in\mathcal{X}_{h,s,o}^m}\frac{S}{N_m(s,a)\vee 1}$$

$$\tag{32}$$

*is a $\xi$-uncertainty quantifier, where $\{\widehat{V}_{h+1}\}_{h\in[H]}$ are obtained in line 8 of Algorithm 1.*

By Theorem 3 and Lemma 7, we have that with probability at least $1 - \xi$

$$\text{SubOpt}_{\mathcal{D}_2}(\widehat{\mu}, s_1)$$

$$\leq \tilde{O}\left(\sum_{h=1}^{H} \mathbb{E}_{\mu^*}\left[H\sqrt{\sum_{h \leq m \leq H}\sum_{s,a}\frac{HS}{N_m(s,a) \vee 1}}\Bigg| s_1\right]\right) + \underbrace{\tilde{O}\left(\sum_{h=1}^{H}\mathbb{E}_{\mu^*}\left[H\phi_h(s,o)|s_1\right]\right)}_{\text{(F. ii)}}$$

$$\underbrace{\phantom{\tilde{O}\left(\sum_{h=1}^{H} \mathbb{E}_{\mu^*}\left[H\sqrt{\sum_{h \leq m \leq H}\sum_{s,a}\frac{HS}{N_m(s,a) \vee 1}}\Bigg| s_1\right]\right)}}_{\text{(F. i)}}$$

Recall that $\mathcal{X}_{h,s,o}^{m}$ is the set of state-action pairs that can be reached at timestep $m \geq h$ by using option $o$ at state $s$ and timestep $h$ without being terminated. Particularly, we have that $\mathcal{X}_{h,s,o}^{h} = \{(s',a')|s'=s, \pi_h^o(a'|s') > 0\}$. For term (F. i), we have that

$$\sqrt{H^3 S} \cdot \tilde{O}\left(\sum_{h=1}^{H}\mathbb{E}_{\mu^*}\left[\sqrt{\sum_{h \leq m \leq H}\sum_{(s,a) \in \mathcal{X}_{h,s,o}^{m}}\frac{1}{N_m(s,a) \vee 1}}\Bigg| s_1\right]\right)$$

$$\leq \sqrt{H^3 S} \cdot \tilde{O}\left(\sum_{h=1}^{H}\sum_{s,o}\sqrt{\frac{\theta_h^{\mu^*}(s,o)}{K}} \cdot \sqrt{\sum_{h \leq m \leq H}\sum_{(s,a) \in \mathcal{X}_{h,s,o}^{m}}\frac{\theta_h^{\mu^*}(s,o)}{d_m^{\rho}(s',a')}}\right) \qquad (33)$$

$$\leq \sqrt{C_2^{\text{option}} H^3 S} \cdot \tilde{O}\left(\sum_{h=1}^{H}\sum_{s,o}\sqrt{\frac{\theta_h^{\mu^*}(s,o)}{K}}\right) \qquad (34)$$

$$\leq \sqrt{C_2^{\text{option}} H^3 S} \cdot \tilde{O}\left(\sqrt{\sum_{h,s,o}\mathbb{I}[\theta_h^{\mu^*}(s,o) > 0]} \cdot \sqrt{\frac{\sum_{h=1}^{H}\sum_{s,o}\theta_h^{\mu^*}(s,o)}{K}}\right)$$

$$\leq \tilde{O}\left(\sqrt{\frac{C_2^{\text{option}} H^3 S Z_{\mathcal{O}}^* \overline{Z}_{\mathcal{O}}^*}{K}}\right) \qquad (35)$$

where Inequality (33) holds by the similar analysis in deriving Inequality (27), Inequality (34) follows from the definition of $C_2^{\text{option}}$ in Equation (9), and the last inequality holds by the definition of $\overline{Z}_{\mathcal{O}}^*$ and $Z_{\mathcal{O}}^*$. We note that since we do not split dataset $\mathcal{D}_2$, a factor of $\sqrt{H}$ is saved in deriving Inequality (33). For term (F. ii) and by the definition of $\phi$ in Equation (32), we have that

$$\tilde{O}\left(\sum_{h=1}^{H}\mathbb{E}_{\mu^*}\left[H\phi_h(s,o)|s_1\right]\right)$$

$$\leq HS \cdot \tilde{O}\left(\mathbb{E}_{\mu^*}\left[\sum_{h \leq m \leq H}\sum_{(s,a) \in \mathcal{X}_{h,s,o}^{m}}\frac{1}{N_m(s,a) \vee 1}\Bigg| s_1\right]\right)$$

$$+ H^2 S^3 \cdot \tilde{O}\left(\sum_{h=1}^{H}\mathbb{E}_{\mu^*}\left[\sum_{h \leq m < t \leq H}\sum_{(s_m,a_m) \in \mathcal{X}_{h,s,o}^{m},(s'_t,a'_t) \in \mathcal{X}_{h,s,o}^{t}}\frac{1}{N_m(s_m,a_m) \vee 1}\Bigg| s_1\right]\right)$$

$$+ H^2 S^3 \cdot \tilde{O}\left(\sum_{h=1}^{H}\mathbb{E}_{\mu^*}\left[\sum_{h \leq m < t \leq H}\sum_{(s_m,a_m) \in \mathcal{X}_{h,s,o}^{m},(s'_t,a'_t) \in \mathcal{X}_{h,s,o}^{t}}\frac{1}{N_t(s'_t,a'_t) \vee 1}\Bigg| s_1\right]\right)$$

$$+ H^2 S^2 \cdot \tilde{O}\left(\mathbb{E}_{\mu^*}\left[\sum_{h \leq m < t \leq H}\sqrt{\sum_{(s_m,a_m) \in \mathcal{X}_{h,s,o}^{m}}\frac{1}{N_m(s_m,a_m) \vee 1}} \cdot \sqrt{\sum_{(s'_t,a'_t) \in \mathcal{X}_{h,s,o}^{t}}\frac{1}{N_t(s'_t,a'_t) \vee 1}}\Bigg| s_1\right]\right)$$

$$\leq H^2 S^3 \cdot \tilde{O}\left(\sum_{h=1}^{H}\sum_{s,o}\theta_h^{\mu^*}(s,o)\sum_{h \leq m < t \leq H}\sum_{(s_m,a_m) \in \mathcal{X}_{h,s,o}^{m},(s'_t,a'_t) \in \mathcal{X}_{h,s,o}^{t}}\frac{1}{Kd_m^{\rho}(s_m,a_m)}\right)$$

$$+ H^2 S^3 \cdot \tilde{O}\left(\sum_{h=1}^{H}\sum_{s,o} \theta_h^{\mu^*}(s,o) \sum_{h \le m < t \le H} \sum_{(s_m,a_m) \in \mathcal{X}_{h,s,o}^m, (s_t',a_t') \in \mathcal{X}_{h,s,o}^t} \frac{1}{K d_t^\rho(s_t',a_t')}\right)$$

$$+ HS \cdot \tilde{O}\left(\sum_{h=1}^{H}\sum_{s,o} \theta_h^{\mu^*}(s,o) \sum_{h \le m \le H} \sum_{(s',a') \in \mathcal{X}_{h,s,o}^m} \frac{1}{K d_m^\rho(s',a')}\right)$$

$$+ H^2 S^2 \cdot \tilde{O}\left(\sum_{h=1}^{H}\sum_{s,o} \theta_h^{\mu^*}(s,o) \sum_{h \le m < t \le H} \sqrt{\sum_{(s_m,a_m) \in \mathcal{X}_{h,s,o}^m} \frac{1}{K d_m^\rho(s_m,a_m)}} \cdot \sqrt{\sum_{(s_t',a_t') \in \mathcal{X}_{h,s,o}^t} \frac{1}{K d_t^\rho(s_t',a_t')}}\right)$$

$$\le C_2^{\text{option}} H^2 S^3 \cdot \tilde{O}\left(\sum_{h=1}^{H}\sum_{s,o} \frac{1}{K} \sum_{h < t \le H, (s_t',a_t') \in \mathcal{X}_{h,s,o}^t} 1\right) + C_2^{\text{option}} H^2 S^3 \cdot \tilde{O}\left(\sum_{h=1}^{H}\sum_{s,o} \frac{1}{K} \sum_{h \le m \le H, (s_m,a_m) \in \mathcal{X}_{h,s,o}^m} 1\right)$$

$$+ \tilde{O}\left(\frac{C_2^{\text{option}} H^2 S^2 O}{K}\right) + \tilde{O}\left(\frac{C_2^{\text{option}} H^4 S^3 O}{K}\right)$$

$$\le \tilde{O}\left(\frac{C_2^{\text{option}} H^4 S^5 AO}{K}\right) \tag{36}$$

where the last-second inequality holds by

$$\sum_{h=1}^{H}\sum_{s,o} \theta_h^{\mu^*}(s,o) \sum_{h \le m < t \le H} \sqrt{\sum_{(s_m,a_m) \in \mathcal{X}_{h,s,o}^m} \frac{1}{K d_m^\rho(s_m,a_m)}} \cdot \sqrt{\sum_{(s_t',a_t') \in \mathcal{X}_{h,s,o}^t} \frac{1}{K d_t^\rho(s_t',a_t')}}$$

$$\le \sum_{h=1}^{H}\sum_{s,o} \frac{1}{K} \sqrt{H \sum_{h \le m \le H} \sum_{(s_m,a_m) \in \mathcal{X}_{h,s,o}^m} \frac{\theta_h^{\mu^*}(s,o)}{d_m^\rho(s_m,a_m)}} \sqrt{H \sum_{h < t \le H} \sum_{(s_t',a_t') \in \mathcal{X}_{h,s,o}^t} \frac{\theta_h^{\mu^*}(s,o)}{d_t^\rho(s_t',a_t')}}$$

$$\le \sum_{h=1}^{H}\sum_{s,o} \frac{H C_2^{\text{option}}}{K} = \frac{H^2 S O C_2^{\text{option}}}{K}$$

Combining Inequalities (35) and (36), when $K \ge \tilde{O}(C_2^{\text{option}} H^5 S^9 A^2 O^2 / (Z_\mathcal{O}^* \overline{Z}_\mathcal{O}^*))$, we conclude the proof. $\qquad\square$

# G Proofs of Auxiliary Lemmas

## G.1 Proof of Lemma 1

*Proof.* We prove this lemma by backward induction. Recall that $\beta_{H+1}^o(s) = 1$ for any $(s,o) \in \mathcal{S} \times \mathcal{O}$. At timestep $H$, for any $s \in \mathcal{S}$, we have that

$$\sum_{\tau \in [H-H+1]} \sum_{s' \in \mathcal{S}} T_H(s'|s,o,\tau) = \sum_{s' \in \mathcal{S}} T_H(s'|s,o,1)$$

$$= \sum_{s' \in \mathcal{S}} \beta_{H+1}^o(s') \sum_{a \in \mathcal{A}} \pi_H^o(a|s) P_H(s'|s,a)$$

$$= \sum_{a \in \mathcal{A}} \pi_H^o(a|s) \sum_{s' \in \mathcal{S}} P_H(s'|s,a) = 1$$

Next, assume that Equation (12) holds for timestep $h + 1$. At timestep $h$, for any $s \in \mathcal{S}$, it holds that

$$\sum_{\tau \in [H-h+1]} \sum_{s' \in \mathcal{S}} T_h(s'|s, o, \tau)$$

$$= \sum_{s' \in \mathcal{S}} T_h(s'|s, o, 1) + \sum_{\tau=2}^{H-h+1} \sum_{s' \in \mathcal{S}} T_h(s'|s, o, \tau)$$

$$= \sum_{s' \in \mathcal{S}} \beta_{h+1}^o(s') \sum_{a \in \mathcal{A}} \pi_h^o(a|s) P_h(s'|s, a)$$

$$+ \sum_{a \in \mathcal{A}} \pi_h^o(a|s) \sum_{s'' \in \mathcal{S}} P_h(s''|s, a) \left(1 - \beta_{h+1}^o(s'')\right) \underbrace{\sum_{\tau=2}^{H-h+1} \sum_{s' \in \mathcal{S}} T_{h+1}(s'|s'', o, \tau - 1)}_{=1}$$

$$= \sum_{s' \in \mathcal{S}} \beta_{h+1}^o(s') \sum_{a \in \mathcal{A}} \pi_h^o(a|s) P_h(s'|s, a) + \sum_{a \in \mathcal{A}} \pi_h^o(a|s) \sum_{s'' \in \mathcal{S}} P_h(s''|s, a) \left(1 - \beta_{h+1}^o(s'')\right)$$

$$= \sum_{s' \in \mathcal{S}} \sum_{a \in \mathcal{A}} \pi_h^o(a|s) P_h(s'|s, a) = 1$$

which concludes the proof. $\qquad\qquad\square$

### G.2 Proof of Lemma 2

*Proof.* Equation (14) can be easily derived by the definitions of $T_h(s'|s, o, 1)$ and $\overline{T}_h(s'|s, o, 1)$. We first prove Equation (16) by backward induction on $t$. When $t = \tau - 1$, by the definition of $\overline{T}_h(s'|s, o, \tau)$ in Equation (13), we have that for any $\tau \geq 2$,

$$\overline{T}_h(s'|s, o, \tau) = \sum_{s'' \in \mathcal{S}} \underbrace{\sum_{a \in \mathcal{A}} \pi_h^o(a|s) P_h(s''|s, a) \left(1 - \beta_{h+1}^o(s'')\right)}_{= \overline{T}_h(s''|s, o, 1)} \overline{T}_{h+1}(s'|s'', o, \tau - 1)$$

$$= \sum_{s'' \in \mathcal{S}} \overline{T}_h(s''|s, o, 1) \overline{T}_{h+1}(s'|s'', o, \tau - 1)$$

Assume that Equation (16) holds for any $\tau \geq 2$ and $2 \leq t + 1 \leq \tau - 1$, we have that

$$\overline{T}_h(s'|s, o, \tau) = \sum_{s'' \in \mathcal{S}} \overline{T}_h(s''|s, o, t + 1) \overline{T}_{h+t+1}(s'|s'', o, \tau - t - 1)$$

$$= \sum_{s''' \in \mathcal{S}} \overline{T}_h(s'''|s, o, t) \underbrace{\sum_{s'' \in \mathcal{S}} \overline{T}_{h+t}(s''|s''', o, 1) \overline{T}_{h+t+1}(s'|s'', o, \tau - t - 1)}_{= \overline{T}_{h+t}(s'|s''', o, \tau - t)}$$

$$= \sum_{s'' \in \mathcal{S}} \overline{T}_h(s''|s, o, t) \overline{T}_{h+t}(s'|s'', o, \tau - t)$$

which concludes the proof of Equation (16). Next, we prove Equation (15) by backward induction on $t$. When $t = \tau - 1$, by the definition of $T_h(s'|s, o, \tau)$ in Equation (11), we have that for any $\tau \geq 2$

$$T_h(s'|s, o, \tau) = \sum_{s'' \in \mathcal{S}} \underbrace{\sum_{a \in \mathcal{A}} \pi_h^o(a|s) P_h(s''|s, a) \left(1 - \beta_{h+1}^o(s'')\right)}_{= \overline{T}_h(s''|s, o, 1)} T_{h+1}(s'|s'', o, \tau - 1)$$

$$= \sum_{s'' \in \mathcal{S}} \overline{T}_h(s''|s, o, 1) T_{h+1}(s'|s'', o, \tau - 1)$$

Assume that Equation (15) holds for any $\tau \geq 2$ and $t + 1 \in [\tau - 1]$, we have that

$$T_h(s'|s, o, \tau) = \sum_{s'' \in \mathcal{S}} \overline{T}_h(s''|s, o, t+1) T_{h+t+1}(s'|s'', o, \tau - t - 1)$$

$$= \sum_{s''' \in \mathcal{S}} \overline{T}_h(s'''|s, o, t) \underbrace{\sum_{s'' \in \mathcal{S}} \overline{T}_{h+t}(s''|s''', o, 1) T_{h+t+1}(s'|s'', o, \tau - t - 1)}_{= T_{h+t}(s'|s''', o, \tau - t)} \quad (37)$$

$$= \sum_{s'' \in \mathcal{S}} \overline{T}_h(s''|s, o, t) T_{h+t}(s'|s'', o, \tau - t)$$

where Equation (37) holds by Equation (16) and we concludes the proof. $\qquad\square$

## G.3 Proof of Lemma 3

*Proof.* Fix any $\mathbf{a}^* \in [B]^{HS}$, by construction of our MDP $M_{\mathbf{a}^*}$, only the good state $s_g$ receives a reward of 1 starting at timestep $h \in \{H + 2, \cdots, 2H + 1\}$. Along each trajectory, there will be exactly one transition from the bandit states $\{s_i\}$ to either the good state $s_g$ or the bad state $s_b$. This transition can happen at timestep $h = H + 1$ but with the same transition probability regardless of the policy. In addition, the state occupancy measure $\theta_h^\mu(s_i) := \sum_o \theta_h^\mu(s_i, o) = \frac{1}{S}(1 - \frac{1}{H})^{h-1} =: \theta_h(s_i)$ (for $h \leq \lfloor z^* \rfloor$) does not depend on the hierarchical policy $\mu$. Note that any option is terminated at any bandit state at the first $\lfloor z^* \rfloor$ timesteps. Therefore, we have that

$$V_{1,M_{\mathbf{a}^*}}^* - V_{1,M_{\mathbf{a}^*}}^{\hat\mu}$$

$$= \sum_{h=1}^{\lfloor z^* \rfloor} \sum_{i=1}^{S} \theta_h(s_i) \cdot \left[ \frac{1}{H}\left(\frac{1}{2} + \tau\right) - \frac{1}{2H} \right] \cdot \mathbb{I}[\hat\mu_h(s_i) \neq o^*] \cdot H$$

$$= \sum_{h=1}^{\lfloor z^* \rfloor} \sum_{i=1}^{S} \frac{1}{S}\left(1 - \frac{1}{H}\right)^{h-1} \tau \cdot \mathbb{I}[\hat\mu_h(s_i) \neq o^*]$$

Taking the expectation with respect to the execution of the behavior policy $\rho$ within the MDP $M_{\mathbf{a}^*}$, we have that

$$\mathbb{E}_{M_{\mathbf{a}^*}}\left[ V_{1,M_{\mathbf{a}^*}}^* - V_{1,M_{\mathbf{a}^*}}^{\hat\mu} \right] = \sum_{h=1}^{\lfloor z^* \rfloor} \sum_{i=1}^{S} \frac{1}{S}\left(1 - \frac{1}{H}\right)^{h-1} \tau \cdot \underbrace{\mathbb{P}_{\mathbf{a}^*}(\hat\mu_h(s_i) \neq o^*)}_{= l_{h,i}(\hat\mu, \mathbf{a}^*)}$$

$$\geq \frac{\tau}{3S} \cdot \sum_{h=1}^{\lfloor z^* \rfloor} \sum_{i=1}^{S} l_{h,i}(\hat\mu, \mathbf{a}^*) = \frac{\tau}{3S} \cdot L(\hat\mu, \mathbf{a}^*)$$

which concludes the proof. $\qquad\square$

## G.4 Proof of Lemma 4

*Proof.* We first provide the extended value difference lemma for options, which generalizes the extended value difference lemma (Cai et al., 2020, Lemma 4.2). (The detailed proof can be found in Appendix G.5.)

**Lemma 8** (Extended value difference for learning with options). *Let $\mu = \{\mu_h\}_{h \in [H]}$ and $\mu' = \{\mu'_h\}_{h \in [H]}$ be any two hierarchical policies and let $\{\widehat{Q}_h : \mathcal{S} \times \mathcal{O} \mapsto \mathbb{R}^+\}_{h \in [H]}$ be any estimated Q-function. For all $h \in [H]$, we define the estimated value function $\widehat{V}_h : \mathcal{S} \mapsto \mathbb{R}$ by setting $\widehat{V}_h(s) = \langle \widehat{Q}_h(s, \cdot), \mu_h(\cdot|s) \rangle_\mathcal{O}$ for all $s \in \mathcal{S}$. For all $s \in \mathcal{S}$, we have*

$$\widehat{V}_1(s) - V_1^{\mu'}(s)$$

$$= \sum_{h=1}^{H} \mathbb{E}_{\mu'}[\langle \widehat{Q}_h(s_h, \cdot), \mu_h(\cdot|s_h) - \mu'_h(\cdot|s_h) \rangle_\mathcal{O}|s_1] - \sum_{h=1}^{H} \mathbb{E}_{\mu'}[\iota_h(s_h, o_h)|s_1] \quad (38)$$

*where $\mathbb{E}_{\mu'}[f(s_h)|s_1] := \sum_{s' \in \mathcal{S}} \theta_h^{\mu'}(s') f(s')$ and $\mathbb{E}_{\mu'}[g(s_h, o_h)] := \sum_{(s', o')} \theta_h^{\mu'}(s', o') g(s', o')$ for any $h \in [H]$ and arbitrary function $f : \mathcal{S} \mapsto \mathbb{R}$ and $g : \mathcal{S} \times \mathcal{O} \mapsto \mathbb{R}$.*

We decompose the suboptimality (3) into two terms.

$$\text{SubOpt}_{\mathcal{D}}(\widehat{\mu}, s_1) = \underbrace{\left(V_1^*(s) - \widehat{V}_1(s)\right)}_{(i)} + \underbrace{\left(\widehat{V}_1(s) - V_1^{\widehat{\mu}}(s)\right)}_{(ii)} \tag{39}$$

**Term (i).** Applying Lemma 8 with $\mu = \widehat{\mu}, \mu' = \mu^*$ and $\{\widehat{Q}_h\}_{h\in[H]}$ being the estimated $Q$-functions constructed by the meta-algorithm (and taking the inverse in both sides), we have that

$$V_1^*(s) - \widehat{V}_1(s) = \sum_{h=1}^{H} \mathbb{E}_{\mu^*}[\langle \widehat{Q}_h(s_h, \cdot), \mu_h^*(\cdot|s_h) - \widehat{\mu}_h(\cdot|s_h)\rangle_{\mathcal{O}}|s_1] + \sum_{h=1}^{H} \mathbb{E}_{\mu^*}[\iota_h(s_h, o_h)|s_1] \tag{40}$$

**Term (ii).** Similarly, applying Lemma 8 with $\mu = \mu' = \widehat{\mu}$ and $\{\widehat{Q}_h\}_{h\in[H]}$ being the estimated $Q$-functions constructed by the meta-algorithm, we have that

$$\widehat{V}_1(s) - V_1^{\widehat{\mu}}(s) = -\sum_{h=1}^{H} \mathbb{E}_{\widehat{\mu}}[\iota_h(s_h, o_h)|s_1] \tag{41}$$

Combining Equation (40) and (41), we decompose the suboptimality (39) as follows

$$\text{SubOpt}_{\mathcal{D}}(\widehat{\mu}, s_1) = -\sum_{h=1}^{H} \mathbb{E}_{\widehat{\mu}}[\iota_h(s_h, o_h)|s_1] + \sum_{h=1}^{H} \mathbb{E}_{\mu^*}[\iota_h(s_h, o_h)|s_1]$$

$$+ \sum_{h=1}^{H} \mathbb{E}_{\mu^*}[\langle \widehat{Q}_h(s_h, \cdot), \mu_h^*(\cdot|s_h) - \widehat{\mu}_h(\cdot|s_h)\rangle_{\mathcal{O}}|s_1]$$

which concludes the proof. □

### G.5 Proof of Lemma 8

*Proof.* By the definition of $\widehat{V}_h(s) = \langle \widehat{Q}_h(s, \cdot), \mu_h(\cdot|s)\rangle_{\mathcal{O}}$ and $V_h^{\mu'} = \langle Q_h^{\mu'}(s, \cdot), \mu_h'(\cdot|s)\rangle_{\mathcal{O}}$, we have that

$$\widehat{V}_h(s) - V_h^{\mu'}(s) = \langle \widehat{Q}_h(s, \cdot), \mu_h(\cdot|s)\rangle_{\mathcal{O}} - \langle Q_h^{\mu'}(s, \cdot), \mu_h'(\cdot|s)\rangle_{\mathcal{O}}$$

$$= \langle \widehat{Q}_h(s, \cdot), \mu_h(\cdot|s) - \mu_h'(\cdot|s)\rangle_{\mathcal{O}} + \langle \widehat{Q}_h(s, \cdot) - Q_h^{\mu'}(s, \cdot), \mu_h'(\cdot|s)\rangle_{\mathcal{O}}$$

By the definition of the model prediction error in Equation (25), we have that

$$Q_h^{\mu'} = U_h + \sum_{\tau=1}^{H-h+1} [T_h V_{h+\tau}^{\mu'}], \quad \widehat{Q}_h = U_h + \sum_{\tau=1}^{H-h+1} [T_h \widehat{V}_{h+\tau}] - \iota_h$$

which implies that

$$\widehat{Q}_h - Q_h^{\mu'} = \sum_{\tau=1}^{H-h+1} [T_h(\widehat{V}_{h+\tau} - V_{h+\tau}^{\mu'})] - \iota_h$$

That is, we have that

$$\widehat{V}_h(s) - V_h^{\mu'}(s) = \sum_{\tau=1}^{H-h+1} \langle [T_h(\widehat{V}_{h+\tau} - V_{h+\tau}^{\mu'})](s, \cdot), \mu_h'(\cdot|s)\rangle_{\mathcal{O}}$$

$$- \langle \iota_h(s, \cdot), \mu_h'(\cdot|s)\rangle_{\mathcal{O}} + \langle \widehat{Q}_h(s, \cdot), \mu_h(\cdot|s) - \mu_h'(\cdot|s)\rangle_{\mathcal{O}} \tag{42}$$

Recall that for any hierarchical policy $\mu$ and episode that starts from state $s_1$ at the first timestep, the state-option occupancy measure $\theta^\mu$ is given by

$$\theta_h^\mu(s, o) = \mu_h(o|s) \sum_{\tau\in[h-1]} \sum_{s'\in\mathcal{S}} \sum_{o'\in\mathcal{O}} \theta_{h-\tau}^\mu(s', o') T_{h-\tau}(s|s', o', \tau)$$

where $\theta_1(s, o) = \mathbb{I}[s = s_1]\mu_1(o|s)$. Let $q_h^\mu(s'|s, \tau) := \sum_{o\in\mathcal{O}} \mu_h(o|s) T_h(s'|s, o, \tau)$ is the probability that the agent selects a new option $o$ according to the hierarchical policy $\mu$ at state $s$ at timestep $h$

(either when $h = 1$ or when the option used at timestep $(h - 1)$ is terminated if $h \geq 2$), uses option $o$ for $\tau$ timesteps, and terminates option $o$ at state $s'$ at timestep $h + \tau$. We have that

$$\theta_h^\mu(s) = \sum_{o \in \mathcal{O}} \theta_h^\mu(s, o) = \sum_{\{(t_i, s_{t_i})\}_{i=1}^j \in \mathcal{L}_{h,s}} \prod_{i=1}^{j-1} q_{t_i}^\mu(s_{t_{i+1}} | s_{t_i}, t_{i+1} - t_i)$$

Here, $\mathcal{L}_{h,s}$ contains any possible sequence of timestep-state pair $\{(t_1, s_{t_1}), (t_2, s_{t_2}), ..., (t_j, s_{t_j})\}$ with a (random) length of $j \leq h$, where $1 = t_1 < t_2 < ... < t_j \leq H$. For such a sequence, the agent selects a new option at state $s_{t_i}$ at timestep $t_i$ (either when $t_i = 1$ or when the option used at timestep $(t_i - 1)$ is terminated if $t_i \geq 2$), uses it for $(t_{i+1} - t_i)$ timesteps, and terminates this option at timestep $t_{i+1}$. It also holds that $(t_1, s_{t_1}) = (1, s_1)$ and $(t_j, s_{t_j}) = (h, s)$. Recursively expanding Equation (42), we obtain that

$$\widehat{V}_1(s) - V_1^{\mu'}(s)$$

$$= \sum_{\tau=1}^{H} \langle [T_1(\widehat{V}_{1+\tau} - V_{1+\tau}^{\mu'})](s, \cdot), \mu_1'(\cdot|s)\rangle_{\mathcal{O}} - \langle \iota_1(s, \cdot), \mu_1'(\cdot|s)\rangle_{\mathcal{O}} + \langle \widehat{Q}_1(s, \cdot), \mu_1(\cdot|s) - \mu_1'(\cdot|s)\rangle_{\mathcal{O}}$$

$$= \sum_{o \in \mathcal{O}} \mu_1'(o|s) \sum_{\tau=1}^{H} \sum_{s' \in \mathcal{S}} T_1(s'|s, o, \tau)\left(\widehat{V}_{1+\tau}(s') - V_{1+\tau}^{\mu'}(s')\right) - \langle \iota_1(s, \cdot), \mu_1'(\cdot|s)\rangle_{\mathcal{O}} + \langle \widehat{Q}_1(s, \cdot), \mu_1(\cdot|s) - \mu_1'(\cdot|s)\rangle_{\mathcal{O}}$$

$$= \sum_{\tau=1}^{H} \sum_{s' \in \mathcal{S}} \underbrace{\left(\sum_{o \in \mathcal{O}} \mu_1'(o|s) T_1(s'|s, o, \tau)\right)}_{:= q_1^{\mu'}(s'|s, \tau)} \left(\sum_{\tau'=1}^{H-\tau} \langle [T_{1+\tau}(\widehat{V}_{1+\tau+\tau'} - V_{1+\tau+\tau'}^{\mu'})](s, \cdot), \mu_{1+\tau}'(\cdot|s')\rangle_{\mathcal{O}} \right.$$

$$\left. - \langle \iota_{1+\tau}(s', \cdot), \mu_{1+\tau}'(\cdot|s')\rangle_{\mathcal{O}} + \langle \widehat{Q}_{1+\tau}(s', \cdot), \mu_{1+\tau}(\cdot|s') - \mu_{1+\tau}'(\cdot|s')\rangle_{\mathcal{O}}\right)$$

$$- \langle \iota_1(s, \cdot), \mu_1'(\cdot|s)\rangle_{\mathcal{O}} + \langle \widehat{Q}_1(s, \cdot), \mu_1(\cdot|s) - \mu_1'(\cdot|s)\rangle_{\mathcal{O}}$$

$$= \sum_{\tau=1}^{H} \sum_{s' \in \mathcal{S}} q_1^{\mu'}(s'|s, \tau) \sum_{\tau'=1}^{H-\tau} \langle [T_{1+\tau}(\widehat{V}_{1+\tau+\tau'} - V_{1+\tau+\tau'}^{\mu'})](s, \cdot), \mu_{1+\tau}'(\cdot|s')\rangle_{\mathcal{O}}$$

$$+ \left(\langle \widehat{Q}_1(s, \cdot), \mu_1(\cdot|s) - \mu_1'(\cdot|s)\rangle_{\mathcal{O}} + \sum_{\tau=1}^{H} \sum_{s' \in \mathcal{S}} q_1^{\mu'}(s'|s, \tau)\langle \widehat{Q}_{1+\tau}(s', \cdot), \mu_{1+\tau}(\cdot|s') - \mu_{1+\tau}'(\cdot|s')\rangle_{\mathcal{O}}\right)$$

$$- \left(\langle \iota_1(s, \cdot), \mu_1'(\cdot|s)\rangle_{\mathcal{O}} + \sum_{\tau=1}^{H} \sum_{s' \in \mathcal{S}} q_1^{\mu'}(s'|s, \tau)\langle \iota_{1+\tau}(s', \cdot), \mu_{1+\tau}'(\cdot|s')\rangle_{\mathcal{O}}\right)$$

$$= \cdots = \sum_{h=1}^{H} \sum_{s \in \mathcal{S}} \theta_h^{\mu'}(s)\langle \widehat{Q}_h(s, \cdot), \mu_h(\cdot|s) - \mu_h'(\cdot|s)\rangle_{\mathcal{O}} - \sum_{h=1}^{H} \sum_{s \in \mathcal{S}} \theta_h^{\mu'}(s)\langle \iota_h(s, \cdot), \mu_h'(\cdot|s)\rangle_{\mathcal{O}}$$

$$= \sum_{h=1}^{H} \mathbb{E}_{\mu'}[\langle \widehat{Q}_h(s_h, \cdot), \mu_h(\cdot|s_h) - \mu_h'(\cdot|s_h)\rangle_{\mathcal{O}}|s_1] - \sum_{h=1}^{H} \mathbb{E}_{\mu'}[\iota_h(s_h, o_h)|s_1]$$

which concludes the proof. □

### G.6 Proof of Lemma 5

*Proof.* We first show that conditioned on the successful event $\mathcal{E}$ defined in Equation (5), the model evaluation error $\{\iota_h\}_{h \in [H]}$ is non-negative. Indeed, by the definition of $\iota_h$ in Equation (25), if $\overline{Q}_h(s, o) < 0$ (which implies $\widehat{Q}_h(s, o) = 0$ by line 6 of Algorithm 1), we have

$$\iota_h(s, o) = U_h(s, o) + \sum_{\tau=1}^{H-h+1} [T_h\widehat{V}_{h+\tau}](s, o) - \widehat{Q}_h(s, o) = U_h(s, o) + \sum_{\tau=1}^{H-h+1} [T_h\widehat{V}_{h+\tau}](s, o) \geq 0$$

Otherwise, if $\overline{Q}_h(s,o) \geq 0$, we have

$$\iota_h(s,o) \geq U_h(s,o) + \sum_{\tau=1}^{H-h+1} [T_h \widehat{V}_{h+\tau}](s,o) - \overline{Q}_h(s,o)$$

$$= U_h(s,o) - \widehat{U}_h(s,o) + \sum_{\tau=1}^{H-h+1} [(T_h - \widehat{T}_h)\widehat{V}_{h+\tau}](s,o) + \Gamma_h(s,o) \geq 0$$

Conditioned on the successful event $\mathcal{E}$, we have that

$$\overline{Q}_h(s,o) = \widehat{U}_h(s,o) + \sum_{\tau=1}^{H-h+1} [\widehat{T}_h \widehat{V}_{h+\tau}](s,o) - \Gamma_h(s,o)$$

$$\leq U_h(s,o) + \sum_{\tau=1}^{H-h+1} [T_h \widehat{V}_{h+\tau}](s,o) \leq H - h + 1$$

Hence, we obtain that

$$\widehat{Q}_h(s,o) = \min\{\overline{Q}_h(s,o), H - h + 1\}^+ = \max\{\overline{Q}_h(s,o), 0\} \geq \overline{Q}_h(s,o)$$

Therefore, we finally derive that

$$\iota_h(s,o) = U_h(s,o) + \sum_{\tau=1}^{H-h+1} [T_h \widehat{V}_{h+\tau}](s,o) - \widehat{Q}_h(s,o)$$

$$\leq U_h(s,o) + \sum_{\tau=1}^{H-h+1} [T_h \widehat{V}_{h+\tau}](s,o) - \overline{Q}_h(s,o)$$

$$= U_h(s,o) - \widehat{U}_h(s,o) + \sum_{\tau=1}^{H-h+1} [(T_h - \widehat{T}_h)\widehat{V}_{h+\tau}](s,o) + \Gamma_h(s,o) \leq 2\Gamma_h(s,o)$$

which concludes the proof. $\qquad\square$

### G.7 Proof of Lemma 6

*Proof.* It suffices to show that for all $(h,s,o) \in [H] \times \mathcal{S} \times \mathcal{O}$, it holds that with probability at least $1 - \xi$

$$\left| \widehat{U}_h(s,o) - U_h(s,o) \right| + \left| \sum_{\tau=1}^{H-h+1} [(\widehat{T}_h - T_h)\widehat{V}_{h+\tau}](s,o) \right| \leq \Gamma_h(s,o) \qquad (43)$$

where $\widehat{T}_h(\cdot,\cdot)$ and $\widehat{U}_h(\cdot,\cdot)$ are given in lines 6 and 7 in Subroutine 2, respectively, $\Gamma_h(\cdot,\cdot)$ is defined in Equation (26), and $\widehat{V}_h(\cdot)$ is defined in line 8 of Algorithm 1. Recall that $n_h(s,o)$ is the number of visits to state-option pair $(s,o)$ at the $h$th timestep in subdataset $\mathcal{D}_{1,h}$. By Hoeffding's Inequality and noting that $U_h(\cdot,\cdot) \in [0, H - h + 1]$, for any $(h,s,o) \in [H] \times \mathcal{S} \times \mathcal{O}$, it holds with probability at least $1 - p$ that

$$\left| \widehat{U}_h(s,o) - U_h(s,o) \right| \leq O\left( \sqrt{\frac{H^2}{n_h(s,o) \vee 1} \log\left(\frac{1}{p}\right)} \right) \qquad (44)$$

By Hoeffding's Inequality and noting that $\widehat{T}_h$ only depends on $\mathcal{D}_{1,h}$ and $\widehat{V}_{h+\tau}$ only depends on $\mathcal{D}_{1,t>h}$, for any $(h,s,o) \in [H] \times \mathcal{S} \times \mathcal{O}$, we have that with probability at least $1 - p$

$$\left| \sum_{\tau=1}^{H-h+1} [(\widehat{T}_h - T_h)\widehat{V}_{h+\tau}](s,o) \right| \leq O\left( \sqrt{\frac{H^2}{n_h(s,o) \vee 1} \log\left(\frac{1}{p}\right)} \right) \qquad (45)$$

Therefore, applying a union bound over $(h,s,o) \in [H] \times \mathcal{S} \times \mathcal{O}$ and letting $p \leftarrow \frac{\xi}{2HSO}$, we conclude the proof. $\qquad\square$

## G.8 Proof of Lemma 7

*Proof.* Our proof relies on the following lemma. (The detailed proof can be found in Appendix G.9.)

**Lemma 9.** *With probability at least $1 - \xi$, for any $(h, s, o) \in [H] \times \mathcal{S} \times \mathcal{O}$, it holds that*

$$\sum_{(s',\tau)\in\mathcal{S}\times[H-h+1]} \left|\widehat{T}_h(s'|s,o,\tau) - T_h(s'|s,o,\tau)\right|$$

$$\leq O\left(\sqrt{\sum_{m=h}^{H}\sum_{(s,a)\in\mathcal{X}_{h,s,o}^m}\frac{HS}{N_m(s,a)\vee 1}\log\left(\frac{HSA}{\xi}\right)} + \phi_h(s,o)\log\left(\frac{HSA}{\xi}\right)\right)$$

*where $\phi_h$ is defined in Equation (32) in the proof.*

Similar to the proof of Lemma 6, it suffices to show that for all $(h, s, o) \in [H] \times \mathcal{S} \times \mathcal{O}$, it holds that with probability at least $1 - \xi$

$$\left|\widehat{U}_h(s,o) - U_h(s,o)\right| + \left|\sum_{\tau=1}^{H-h+1}[(\widehat{T}_h - T_h)\widehat{V}_{h+\tau}](s,o)\right| \leq \Gamma_h(s,o) \tag{46}$$

where $\widehat{T}_h(\cdot, \cdot)$ and $\widehat{U}_h(\cdot, \cdot)$ are constructed by replacing $P_h$, $r_h$, and $\overline{T}_h$ with their empirical counterparts in Equation (11) and Equation (17), respectively, $\Gamma_h(\cdot, \cdot)$ is defined in Equation (31), and $\widehat{V}_h(\cdot)$ is defined in line 8 of Algorithm 1. First, recall that

$$U_h(s,o) = \sum_{a\in\mathcal{A}}\pi_h^o(a|s)r_h(s,a) + \sum_{l\in[H-h]}\sum_{s'\in\mathcal{S}}\sum_{a'\in\mathcal{A}}\overline{T}_h(s'|s,o,l)\pi_{h+l}^o(a'|s')r_{h+l}(s',a')$$

Therefore, we have that with probability at least $1 - \xi$,

$$\left|\widehat{U}_h(s,o) - U_h(s,o)\right|$$

$$\leq \sum_{a\in\mathcal{A}}\pi_h^o(a|s)|r_h(s,a) - \widehat{r}_h(s,a)|$$

$$+ \sum_{l\in[H-h]}\sum_{s'\in\mathcal{S}}\sum_{a'\in\mathcal{A}}\pi_{h+l}^o(a'|s')\left|\widehat{\overline{T}}_h(s'|s,o,l)\widehat{r}_{h+l}(s',a') - \overline{T}_h(s'|s,o,l)r_{h+l}(s',a')\right|$$

$$\leq \sum_{a\in\mathcal{A}}\pi_h^o(a|s)|r_h(s,a) - \widehat{r}_h(s,a)| + \sum_{l\in[H-h]}\sum_{(s',a')\in\mathcal{X}_{h,s,o}^l}\sum_{a'\in\mathcal{A}}\pi_{h+l}^o(a'|s')|r_{h+l}(s',a') - \widehat{r}_{h+l}(s',a')|$$

$$\leq O\left(\sum_{m=h}^{H}\sum_{(s,a)\in\mathcal{X}_{h,s,o}^m}\pi_m^o(a|s)\sqrt{\frac{1}{N_m(s,a)\vee 1}\log\left(\frac{HSA}{\xi}\right)}\right)$$

$$\leq O\left(\sqrt{\left(\sum_{m=h}^{H}\sum_{(s,a)\in\mathcal{X}_{h,s,o}^m}(\pi_m^o(a|s))^2\right)\left(\sum_{m=h}^{H}\sum_{(s,a)\in\mathcal{X}_{h,s,o}^m}\frac{1}{N_m(s,a)\vee 1}\log\left(\frac{HSA}{\xi}\right)\right)}\right)$$

$$\leq O\left(\sqrt{\sum_{m=h}^{H}\sum_{(s,a)\in\mathcal{X}_{h,s,o}^m}\frac{HS}{N_m(s,a)\vee 1}\log\left(\frac{HSA}{\xi}\right)}\right)$$

where the last second inequality holds by Cauchy-Schwarz inequality and the last inequality follows from the fact that $\sum_{(s,a)\in\mathcal{X}_{h,s,o}^m}(\pi_m^o(a|s))^2 \leq \sum_{(s,a)\in\mathcal{X}_{h,s,o}^m}\pi_m^o(a|s) \leq S$. In addition, by Lemma 9, we have that

$$\left|\sum_{\tau=1}^{H-h+1}[(\widehat{T}_h - T_h)\widehat{V}_{h+\tau}](s,o)\right|$$

$$\leq H \sum_{\tau\in[H-h+1]} \sum_{s'\in\mathcal{S}} \left|\widehat{T}_h(s'|s,o,\tau) - T_h(s'|s,o,\tau)\right|$$

$$\leq O\left( H\sqrt{\sum_{m=h}^{H} \sum_{(s,a)\in\mathcal{X}_{h,s,o}^m} \frac{HS}{N_m(s,a)\vee 1} \log\left(\frac{HSA}{\xi}\right)} + H\phi_h(s,o)\log\left(\frac{HSA}{\xi}\right)\right)$$

which concludes the proof. $\qquad\square$

### G.9 Proof of Lemma 9

*Proof.* We adopt a similar analysis to the proof of (Jin et al., 2020, Lemma 4). By the construction of $\widehat{T}_h(s'|s,o,\tau)$ in Line 12 of Subroutine 3, we have that

$$\widehat{T}_h(s'|s,o,\tau)$$
$$= \sum_{a\in\mathcal{A}} \pi_h^o(a|s) \sum_{s''\in\mathcal{S}} \widehat{P}_{h+\tau-1}(s''|s,a)\left(1-\beta_{h+1}^o(s'')\right)\widehat{T}_{h+1}(s'|s'',o,\tau-1)$$
$$= \beta_{h+\tau}^o(s') \sum_{\{s_t,a_t\}_{t=h}^{h+\tau-1}} \prod_{t=h}^{h+\tau-1} \left(1-\beta_t^o(s_t)\right)\pi_h^o(a_t|s_t) \prod_{t=h}^{h+\tau-1} \widehat{P}_t(s_{t+1}|s_t,a_t)$$

where $s_h = s, s_{h+\tau} = s'$. Define $\eta_h^o(a|s) := (1-\beta_h^o(s))\pi_h^o(a|s)$. Hence, we have

$$\widehat{T}_h(s'|s,o,\tau) - T_h(s'|s,o,\tau)$$
$$= \beta_{h+\tau}^o(s') \sum_{\{s_t,a_t\}_{t=h}^{h+\tau-1}} \prod_{t=h}^{h+\tau-1} \eta_t^o(a_t|s_t)\underbrace{\left(\prod_{t=h}^{h+\tau-1}\widehat{P}_t(s_{t+1}|s_t,a_t) - \prod_{t=h}^{h+\tau-1}P_t(s_{t+1}|s_t,a_t)\right)}_{(*)} \quad (47)$$

Consider any such trajectory $\{s_t,a_t\}_{t=h}^{h+\tau-1}$. To bound term (*) by the error in estimating the one-step transition kernel, i.e., $|P_h(s'|s,a) - \widehat{P}_h(s'|s,a)|$, we add and subtract $\tau-1$ terms and rewrite it as

$$(*) = \underbrace{\prod_{t=h}^{h+\tau-1}\widehat{P}_t(s_{t+1}|s_t,a_t)}_{(i)} - \prod_{t=h}^{h+\tau-1}P_t(s_{t+1}|s_t,a_t) \pm \underbrace{\sum_{m=h+1}^{h+\tau-1}\prod_{t=h}^{m-1}P_t(s_{t+1}|s_t,a_t)\prod_{t=m}^{h+\tau-1}\widehat{P}_t(s_{t+1}|s_t,a_t)}_{(ii)}$$

$$= \underbrace{(\widehat{P}_h(s_{h+1}|s_h,a_h) \pm P_h(s_{h+1}|s_h,a_h))\prod_{t=h+1}^{h+\tau-1}\widehat{P}_t(s_{t+1}|s_t,a_t)}_{(i)}$$

$$\underbrace{\pm P_h(s_{h+1}|s_h,a_h)\prod_{t=h+1}^{h+\tau-1}\widehat{P}_t(s_{t+1}|s_t,a_t) \pm \sum_{m=h+2}^{h+\tau-1}\prod_{t=h}^{m-1}P_t(s_{t+1}|s_t,a_t)\prod_{t=m}^{h+\tau-1}\widehat{P}_t(s_{t+1}|s_t,a_t)}_{(ii)}$$

$$- \prod_{t=h}^{h+\tau-1}P_t(s_{t+1}|s_t,a_t)$$

$$= (\widehat{P}_h(s_{h+1}|s_h,a_h) - P_h(s_{h+1}|s_h,a_h))\prod_{t=h+1}^{h+\tau-1}\widehat{P}_t(s_{t+1}|s_t,a_t)$$

$$+ P_h(s_{h+1}|s_h,a_h)\left(\widehat{P}_{h+1}(s_{h+2}|s_{h+1},a_{h+1}) \pm P_{h+1}(s_{h+2}|s_{h+1},a_{h+1})\right)\prod_{t=h+2}^{h+\tau-1}\widehat{P}_t(s_{t+1}|s_t,a_t)$$

$$\pm \sum_{m=h+2}^{h+\tau-1}\prod_{t=h}^{m-1}P_t(s_{t+1}|s_t,a_t)\prod_{t=m}^{h+\tau-1}\widehat{P}_t(s_{t+1}|s_t,a_t) - \prod_{t=h}^{h+\tau-1}P_t(s_{t+1}|s_t,a_t)$$

$$= \cdots$$

$$= \sum_{m=h}^{h+\tau-1} \left( \widehat{P}_m(s_{m+1}|s_m, a_m) - P_m(s_{m+1}|s_m, a_m) \right) \prod_{t=h}^{m-1} P_t(s_{t+1}|s_t, a_t) \prod_{t=m+1}^{h+\tau-1} \widehat{P}_t(s_{t+1}|s_t, a_t)$$
(48)

The following lemma shows the error of the empirical transition kernel.

**Lemma 10.** *By the empirical Bernstein Inequality, with probability at least $1 - 4p$, it holds that for any $(h, s, a, s') \in [H] \times \mathcal{S}^2 \times \mathcal{A}$*

$$\left| P_h(s'|s, a) - \widehat{P}_h(s'|s, a) \right| \leq \sqrt{\frac{2\widehat{P}_h(s'|s, a)(1 - \widehat{P}_h(s'|s, a))}{(N_h(s, a) - 1) \vee 1} \log\left(\frac{1}{p}\right)} + \frac{7\log\left(\frac{1}{p}\right)}{3((N_h(s, a) - 1) \vee 1)}$$

*which implies*

$$\left| P_h(s'|s, a) - \widehat{P}_h(s'|s, a) \right| \leq \epsilon_h(s'|s, a)$$
(49)

*for any $(h, s, a) \in [H] \times \mathcal{S} \times \mathcal{A}$, where $N_h(s, a)$ is the number of visits to the state-action pair $(s, a)$ at the $h$th timestep in Dataset $\mathcal{D}_2$ and*

$$\epsilon_h(s'|s, a) := \min\left\{ 1, O\left( \sqrt{\frac{P_h(s'|s, a)}{N_h(s, a) \vee 1} \log\left(\frac{HSA}{p}\right)} + \frac{\log\left(\frac{HSA}{p}\right)}{N_h(s, a) \vee 1} \right) \right\}$$
(50)

*Proof.* See the proof of (Jin et al., 2020, Lemmas 2 and 8). □

Combining Inequalities (47), (48), and (50), we further derive that

$$\left| \widehat{T}_h(s'|s, o, \tau) - T_h(s'|s, o, \tau) \right|$$

$$\leq \beta_{h+\tau}^o(s') \sum_{\{s_t, a_t\}_{t=h}^{h+\tau-1}} \prod_{t=h}^{h+\tau-1} \eta_t^o(a_t|s_t) \sum_{m=h}^{h+\tau-1} \epsilon_m(s_{m+1}|s_m, a_m) \prod_{t=h}^{m-1} P_t(s_{t+1}|s_t, a_t) \prod_{t=m+1}^{h+\tau-1} \widehat{P}_t(s_{t+1}|s_t, a_t)$$

$$= \sum_{m=h}^{h+\tau-1} \sum_{\{s_t, a_t\}_{t=h}^{h+\tau-1}} \epsilon_m(s_{m+1}|s_m, a_m) \left( \eta_m^o(a_m|s_m) \prod_{t=h}^{m-1} \eta_t^o(a_t|s_t) P_t(s_{t+1}|s_t, a_t) \right)$$

$$\cdot \left( \beta_{h+\tau}^o(s') \prod_{t=m+1}^{h+\tau-1} \eta_t^o(a_t|s_t) \widehat{P}_t(s_{t+1}|s_t, a_t) \right)$$

$$= \sum_{m=h}^{h+\tau-1} \sum_{s_m, a_m, s_{m+1}} \epsilon_m(s_{m+1}|s_m, a_m) \left( \sum_{\{s_t, a_t\}_{t=h}^{m-1}} \eta_m^o(a_m|s_m) \prod_{t=h}^{m-1} \eta_t^o(a_t|s_t) P_t(s_{t+1}|s_t, a_t) \right)$$

$$\cdot \left( \sum_{a_{m+1}} \sum_{\{s_t, a_t\}_{t=m+2}^{h+\tau-1}} \beta_{h+\tau}^o(s') \prod_{t=m+1}^{h+\tau-1} \eta_t^o(a_t|s_t) \widehat{P}_t(s_{t+1}|s_t, a_t) \right)$$

$$= \sum_{m=h}^{h+\tau-1} \sum_{s_m, a_m, s_{m+1}} \epsilon_m(s_{m+1}|s_m, a_m) \overline{T}_h(s_m|s, o, m - h) \widehat{T}_h(s'|s_{m+1}, o, h + \tau - (m+1))$$
(51)

Similarly, we have that

$$|\widehat{T}_h(s'|s_{m+1}, o, h + \tau - (m+1)) - T_h(s'|s_{m+1}, o, h + \tau - (m+1))|$$

$$\leq \sum_{t=m+1}^{h+\tau-1} \sum_{s_t', a_t', s_{t+1}'} \epsilon_t(s_{t+1}'|s_t', a_t') \overline{T}_h(s_t'|s_{m+1}, o, t - (m+1)) \widehat{T}_h(s'|s_t', o, h + \tau - t)$$

$$\leq \beta^o_{h+\tau}(s') \sum_{t=m+1}^{h+\tau-1} \sum_{s'_t,a'_t,s'_{t+1}} \epsilon_t(s'_{t+1}|s'_t,a'_t)\overline{T}_h(s'_t|s_{m+1},o,t-(m+1)) \tag{52}$$

for any $m \in \{h,\cdots,h+\tau-1\}$, where the last inequality holds by $\widehat{T}_h(s'|s'_t,o,h+\tau-t) \leq \beta^o_{h+\tau}(s')$.
Let $w_m := (s_m,a_m,s_{m+1})$. Combining Inequality (51) and (52), we derive that

$$\sum_{(s',\tau)\in\mathcal{S}\times[H-h+1]} \left|\widehat{T}_h(s'|s,o,\tau) - T_h(s'|s,o,\tau)\right|$$

$$\leq \sum_{s',\tau} \sum_{m=h}^{h+\tau-1} \sum_{w_m} \epsilon_m(s_{m+1}|s_m,a_m)\overline{T}_h(s_m,a_m|s,o,m-h)T_h(s'|s_{m+1},o,h+\tau-(m+1))$$

$$+ \sum_{s',\tau} \sum_{m=h}^{h+\tau-1} \sum_{w_m} \epsilon_m(s_{m+1}|s_m,a_m)\overline{T}_h(s_m,a_m|s,o,m-h)$$

$$\cdot \left(\beta^o_{h+\tau}(s') \sum_{t=m+1}^{h+\tau-1} \sum_{w'_t} \epsilon(s'_{t+1}|s'_t,a'_t)\overline{T}_h(s'_t,a'_t|s_{m+1},o,t-(m+1))\right)$$

$$= \sum_{m=h}^{H} \sum_{w_m} \epsilon_m(s_{m+1}|s_m,a_m)\overline{T}_h(s_m,a_m|s,o,m-h)\underbrace{\sum_{s',\tau} T_h(s'|s_{m+1},o,h+\tau-(m+1))}_{=\,1\text{ by Lemma 1}}$$

$$+ \sum_{m=h}^{H} \sum_{w_m} \sum_{t=m+1}^{d-1} \sum_{w'_t} \epsilon_m(s_{m+1}|s_m,a_m)\overline{T}_h(s_m,a_m|s,o,m-h)\epsilon(s'_{t+1}|s'_t,a'_t)\overline{T}_h(s'_t,a'_t|s_{m+1},o,t-(m+1))$$

$$\cdot \underbrace{\left(\sum_{s',\tau} \beta^o_{h+\tau}(s')\right)}_{\leq\,HS}$$

$$\leq \underbrace{\sum_{m=h}^{H} \sum_{w_m} \epsilon_m(s_{m+1}|s_m,a_m)\overline{T}_h(s_m,a_m|s,o,m-h)}_{B_1}$$

$$+ HS \underbrace{\sum_{h\leq m<t\leq H} \sum_{w_m,w'_t} \epsilon_m(s_{m+1}|s_m,a_m)\overline{T}_h(s_m,a_m|s,o,m-h)\epsilon_t(s'_{t+1}|s'_t,a'_t)\overline{T}_h(s'_t,a'_t|s_{m+1},o,t-(m+1))}_{B_2}$$

$$\tag{53}$$

**Step 1. Bounding term $B_1$.** Recall that from Equation (50)

$$\epsilon_h(s'|s,a) = \min\left\{1, O\left(\sqrt{\frac{P_h(s'|s,a)}{N_h(s,a)\vee 1}\log\left(\frac{HSA}{p}\right)} + \frac{\log\left(\frac{HSA}{p}\right)}{N_h(s,a)\vee 1}\right)\right\}$$

We have that

$$B_1 = \sum_{m=h}^{H} \sum_{w_m} \epsilon_m(s_{m+1}|s_m,a_m)\overline{T}_h(s_m,a_m|s,o,m-h)$$

$$= O\left(\sum_{m=h}^{H} \sum_{w_m} \overline{T}_h(s_m,a_m|s,o,m-h)\sqrt{\frac{P_m(s_{m+1}|s_m,a_m)}{N_m(s_m,a_m)\vee 1}\log\left(\frac{HSA}{p}\right)}\right.$$

$$\left.+ \sum_{m=h}^{H} \sum_{w_m} \frac{\overline{T}_h(s_m,a_m|s,o,m-h)}{N_m(s_m,a_m)\vee 1}\log\left(\frac{HSA}{p}\right)\right)$$

For the first term, applying Cauchy-Schwarz inequality yields

$$\sum_{m=h}^{H} \sum_{w_m} \overline{T}_h(s_m, a_m | s, o, m-h) \sqrt{\frac{P_m(s_{m+1}|s_m, a_m)}{N_m(s_m, a_m) \vee 1} \log\left(\frac{HSA}{p}\right)}$$

$$\leq \sum_{m=h}^{H} \sum_{(s_m, a_m) \in \mathcal{X}_{h,s,o}^m} \overline{T}_h(s_m, a_m | s, o, m-h) \sum_{s_{m+1} \in \mathcal{S}} \sqrt{\frac{P_m(s_{m+1}|s_m, a_m)}{N_m(s_m, a_m) \vee 1} \log\left(\frac{HSA}{p}\right)}$$

$$\leq \sum_{m=h}^{H} \sum_{(s_m, a_m) \in \mathcal{X}_{h,s,o}^m} \overline{T}_h(s_m, a_m | s, o, m-h) \sqrt{\left(\sum_{s_{m+1} \in \mathcal{S}} 1\right) \left(\frac{\sum_{s_{m+1} \in \mathcal{S}} P_m(s_{m+1}|s_m, a_m)}{N_m(s_m, a_m) \vee 1} \log\left(\frac{HSA}{p}\right)\right)}$$

$$= \sum_{m=h}^{H} \sum_{(s_m, a_m) \in \mathcal{X}_{h,s,o}^m} \overline{T}_h(s_m, a_m | s, o, m-h) \sqrt{\frac{S}{N_m(s_m, a_m) \vee 1} \log\left(\frac{HSA}{p}\right)}$$

$$\leq \sqrt{\left(\sum_{m=h}^{H} \sum_{(s_m, a_m) \in \mathcal{X}_{h,s,o}^m} (\overline{T}_h(s_m, a_m | s, o, m-h))^2\right) \left(\sum_{m=h}^{H} \sum_{(s_m, a_m) \in \mathcal{X}_{h,s,o}^m} \frac{S}{N_m(s_m, a_m) \vee 1} \log\left(\frac{HSA}{p}\right)\right)}$$

$$\leq \sqrt{\sum_{m=h}^{H} \sum_{(s_m, a_m) \in \mathcal{X}_{h,s,o}^m} \frac{HS}{N_m(s_m, a_m) \vee 1} \log\left(\frac{HSA}{p}\right)}$$

where the last inequality holds by $\sum_{(s_m, a_m) \in \mathcal{X}_{h,s,o}^m} (\overline{T}_h(s_m, a_m | s, o, m-h))^2 \leq \sum_{(s_m, a_m) \in \mathcal{X}_{h,s,o}^m} \overline{T}_h(s_m, a_m | s, o, m-h) \leq 1$, i.e., it is the probability that the agent does not terminate the option at timestep $m$. Hence, we obtain that

$$B_1 \leq O\left(\sqrt{\sum_{m=h}^{H} \sum_{(s_m, a_m) \in \mathcal{X}_{h,s,o}^m} \frac{HS}{N_m(s_m, a_m) \vee 1} \log\left(\frac{HSA}{p}\right)}\right.$$
$$\left. + \sum_{m=h}^{H} \sum_{(s_m, a_m) \in \mathcal{X}_{h,s,o}^m} \frac{S}{N_m(s_m, a_m) \vee 1} \log\left(\frac{HSA}{p}\right)\right)$$

$$(54)$$

**Step 2. Bounding term $B_2$.** Since $\epsilon_h, \overline{T}_h \leq 1$, we have that

$$B_2 \leq \sum_{h \leq m < t \leq H} \sum_{w_m, w_t'} \left(\sqrt{\frac{P_m(s_{m+1}|s_m, a_m)}{N_m(s_m, a_m) \vee 1} \log\left(\frac{HSA}{p}\right)} \overline{T}_h(s_m, a_m | s, o, m-h)\right.$$

$$\left. \cdot \sqrt{\frac{P_t(s_{t+1}|s_t, a_t)}{N_t(s_t', a_t') \vee 1} \log\left(\frac{HSA}{p}\right)} \overline{T}_h(s_t', a_t' | s_{m+1}, o, t-(m+1))\right)$$

$$+ \sum_{h \leq m < t \leq H} \sum_{w_m, w_t'} \frac{\overline{T}_h(s_m, a_m | s, o, m-h)}{N_m(s_m, a_m) \vee 1} \log\left(\frac{HSA}{p}\right)$$

$$+ \sum_{h \leq m < t \leq H} \sum_{w_m, w_t'} \frac{\overline{T}_h(s_t', a_t' | s_{m+1}, o, t-(m+1))}{N_t(s_t', a_t') \vee 1} \log\left(\frac{HSA}{p}\right)$$

$$(55)$$

Applying the Cauchy-Schwartz inequality, the first term of Inequality (55) can be written as

$$\sum_{h \leq m < t \leq H} \sum_{w_m, w_t'} \log\left(\frac{HSA}{p}\right) \sqrt{\frac{\overline{T}_h(s_m, a_m | s, o, m-h) P_t(s_{t+1}'|s_t', a_t') \overline{T}_h(s_t', a_t' | s_{m+1}, o, t-(m+1))}{N_m(s_m, a_m) \vee 1}}$$

$$\cdot \sqrt{\frac{\overline{T}_h(s_m, a_m | s, o, m-h) P_m(s_{m+1} | s_m, a_m) \overline{T}_h(s_t', a_t' | s_{m+1}, o, t-(m+1))}{N_t(s_t', a_t') \vee 1}}$$

$$\leq \sum_{h \leq m < t \leq H} \log\left(\frac{HSA}{p}\right) \sqrt{\sum_{w_m, w_t'} \frac{\overline{T}_h(s_m, a_m | s, o, m-h) P_t(s_{t+1}' | s_t', a_t') \overline{T}_h(s_t', a_t' | s_{m+1}, o, t-(m+1))}{N_m(s_m, a_m) \vee 1}}$$

$$\cdot \sqrt{\sum_{w_m, w_t'} \frac{\overline{T}_h(s_m, a_m | s, o, m-h) P_m(s_{m+1} | s_m, a_m) \overline{T}_h(s_t', a_t' | s_{m+1}, o, t-(m+1))}{N_t(s_t', a_t') \vee 1}}$$

$$\leq \sum_{h \leq m < t \leq H} \log\left(\frac{HSA}{p}\right) \sqrt{S \sum_{(s_m, a_m) \in \mathcal{X}_{h,s,o}^m} \frac{\overline{T}_h(s_m, a_m | s, o, m-h)}{N_m(s_m, a_m) \vee 1}} \cdot \sqrt{S \sum_{(s_t', a_t') \in \mathcal{X}_{h,s,o}^t} \frac{\overline{T}_h(s_t', a_t' | s, o, t-h)}{N_t(s_t', a_t') \vee 1}}$$

$$(56)$$

where the last inequality holds by

$$\sum_{w_m, w_t'} \frac{\overline{T}_h(s_m, a_m | s, o, m-h) P_t(s_{t+1}' | s_t', a_t') \overline{T}_h(s_t', a_t' | s_{m+1}, o, t-(m+1))}{N_m(s_m, a_m) \vee 1}$$

$$= \sum_{w_m} \frac{\overline{T}_h(s_m, a_m | s, o, m-h)}{N_m(s_m, a_m) \vee 1} \underbrace{\sum_{s_t', a_t'} \overline{T}_h(s_t', a_t' | s_{m+1}, o, t-(m+1)) \underbrace{\sum_{s_{t+1}'} P_t(s_{t+1}' | s_t', a_t')}_{= 1}}_{\leq 1}$$

$$\leq S \sum_{(s_m, a_m) \in \mathcal{X}_{h,s,o}^m} \frac{\overline{T}_h(s_m, a_m | s, o, m-h)}{N_m(s_m, a_m) \vee 1}$$

and

$$\sum_{w_m, w_t'} \frac{\overline{T}_h(s_m, a_m | s, o, m-h) P_m(s_{m+1} | s_m, a_m) \overline{T}_h(s_t', a_t' | s_{m+1}, o, t-(m+1))}{N_t(s_t', a_t') \vee 1}$$

$$\leq S \sum_{(s_t', a_t') \in \mathcal{X}_{h,s,o}^t} \frac{\overline{T}_h(s_t', a_t' | s, o, t-h)}{N_t(s_t', a_t') \vee 1}$$

by the same analysis.

In addition, the second term of Inequality (55) can be bounded by

$$\sum_{h \leq m < t \leq H} \sum_{w_m, w_t'} \frac{\overline{T}_h(s_m, a_m | s, o, m-h)}{N_m(s_m, a_m) \vee 1} \log\left(\frac{HSA}{p}\right)$$

$$\leq \sum_{h \leq m < t \leq H} \sum_{w_m, w_t'} \frac{1}{N_m(s_m, a_m) \vee 1} \log\left(\frac{HSA}{p}\right)$$

$$= S^2 \sum_{h \leq m < t \leq H} \sum_{(s_m, a_m) \in \mathcal{X}_{h,s,o}^m, (s_t', a_t') \in \mathcal{X}_{h,s,o}^t} \frac{1}{N_m(s_m, a_m) \vee 1} \log\left(\frac{HSA}{p}\right) \qquad (57)$$

By the exact analysis, the third term of Inequality (55) can be bounded by

$$\sum_{h \leq m < t \leq H} \sum_{w_m, w_t'} \frac{\overline{T}_h(s_t', a_t' | s_{m+1}, o, t-(m+1))}{N_t(s_t', a_t') \vee 1} \log\left(\frac{1}{p}\right)$$

$$\leq S^2 \sum_{h \leq m < t \leq H} \sum_{(s_m, a_m) \in \mathcal{X}_{h,s,o}^m, (s_t', a_t') \in \mathcal{X}_{h,s,o}^t} \frac{1}{N_t(s_t', a_t') \vee 1} \log\left(\frac{HSA}{p}\right) \qquad (58)$$

Combining Inequalities (56), (57), and (58), we have that

$$
\begin{aligned}
B_2 \leq & S \sum_{h \leq m < t \leq H} \sqrt{\sum_{(s_m, a_m) \in \mathcal{X}_{h,s,o}^m} \frac{1}{N_m(s_m, a_m) \vee 1}} \cdot \sqrt{\sum_{(s_t', a_t') \in \mathcal{X}_{h,s,o}^t} \frac{1}{N_t(s_t', a_t') \vee 1}} \\
& + S^2 \sum_{h \leq m < t \leq H} \sum_{(s_m, a_m) \in \mathcal{X}_{h,s,o}^m, (s_t', a_t') \in \mathcal{X}_{h,s,o}^t} \left( \frac{1}{N_m(s_m, a_m) \vee 1} + \frac{1}{N_t(s_t', a_t') \vee 1} \right)
\end{aligned}
\tag{59}
$$

**Step 3. Putting all together.** Therefore, combining Inequalities (53), (54), and (59), we derive that

$$
\begin{aligned}
& \sum_{(s', \tau) \in \mathcal{S} \times [H-h+1]} \left| \widehat{T}_h(s'|s, o, \tau) - T_h(s'|s, o, \tau) \right| \\
\leq & B_1 + HS \cdot B_2 \\
\leq & \sqrt{\sum_{m=h}^{H} \sum_{(s_m, a_m) \in \mathcal{X}_{h,s,o}^m} \frac{HS}{N_m(s_m, a_m) \vee 1} \log\left(\frac{HSA}{p}\right) + \phi_h(s, o) \log\left(\frac{HSA}{p}\right)}
\end{aligned}
$$

where $\phi_h$ is defined in Equation (32). $\qquad \square$

# H  Counterexample

*Example* 1. We consider an episodic MDP with the following structures: (1) $P_{H-1}(x|s, a) = 1$ for some $x \in \mathcal{S}$ and any $(s, a) \in \mathcal{S} \times \mathcal{A}$, i.e., the agent is guaranteed to arrive state $x$ at the $H$th timestep using any hierarchical policy, (2) the set of actions $\mathcal{A} = \{a^1, a^2\}$, which satisfies that $r_H(x, a^1) = 1$ and $r_H(x, a^2) = 0$, (3) the set of options $\mathcal{O} = \{o_1, o_2\}$, which satisfies that: (3.i) $o_1$ and $o_2$ are exactly the same before timestep $H$, i.e., $\pi_h^{o_1}(a|s) = \pi_h^{o_2}(a|s), \beta_h^{o_1}(s) = \beta_h^{o_2}(s)$ for any $(h, s, a) \in [H-1] \times \mathcal{S} \times \mathcal{A}$, (3.ii) at state $x$ at timestep $H$, option $o_1$ always takes action $a^1$ while option $o_2$ takes action $a^2$ with probability $\epsilon \in (0, 1)$, i.e., $\pi_H^{o_1}(a^1|x) = 1$, $\pi_H^{o_2}(a^1|x) = 1 - \epsilon$, $\pi_H^{o_2}(a^2|x) = \epsilon$, (3.iii) the agent guarantees to terminate option $o_{h-1}$ and select a new option at timestep $H$, i.e., $\beta_H^{o_1}(x) = \beta_H^{o_2}(x) = 1$. It can be easily seen that $\mu_H^*(x) = o_1$. If the hierarchical behavior policy $\rho$ to collect the dataset $\mathcal{D}_1$ always selects $o_2$ in the $H$th timestep, i.e., $\rho_H(x) = o_2$, then $\widehat{\mu}_H(x) = o_2$ since no information of $o_1$ is provided. Therefore, it holds that $C_1^{\text{option}} = \infty$ while $C_2^{\text{option}} < \infty$.