# OpenReview forum: "Provably (More) Sample-Efficient Offline RL with Options"
_NeurIPS.cc/2023/Conference — NeurIPS 2023 poster_

### Official Review · Reviewer_V1tj · 2023-07-05

**Soundness:** 2 fair
**Presentation:** 3 good
**Contribution:** 3 good
**Rating:** 7
**Confidence:** 3

**Summary:**

The paper investigates the sample complexity of offline reinforcement learning (RL) with options. An information-theoretic lower bound for the suboptimality of the expected return of the policy over options that is learned offline is derived (by constructing hard MDP instances). The main contribution of the paper is the PEVIO algorithm, a version of value iteration for offline learning with options, and its theoretical analysis. The main idea behind PEVIO (borrowed from the PEVI algorithm for offline RL with primitive actions) is to be pessimistic in the face of uncertainty arising from the finite, fixed dataset. This pessimism is implemented by subtracting a penalty function from the value estimates which bounds the error in the estimates for the option transition function and option utilities. Suboptimality bounds are derived for two versions of PEVIO, intended for learning from state-option-utility and state-action-reward triplets. Depending on the frequency of option switches and the optimality of the option set, these bounds show that offline learning with options can be more sample efficient compared to learning with primitive actions.

**Strengths:**

As offline RL is highly relevant for many applications and hierarchical RL with the options framework promises to speed up learning, a theoretical analysis of the sample complexity offline RL with options is of interest to the RL community. To my knowledge, the results in the paper are novel and relevant. The paper is comprehensive in the sense that it presents both an information-theoretic lower bound and a concrete algorithm with suboptimality bounds. The advantages and disadvantages of storing state-option or state-action transitions are discussed which are relevant for practical applications. The majority of the paper is well written and the presentation is well structured.

**Weaknesses:**

From the perspective of somebody mainly interested in practical algorithms and applications, PEVIO seems to be quite far away from an algorithm that would perform well in practice. In particular the data splitting technique discussed in line 214 seems to effectively throw away a large fraction $\frac{H – 1}{H}$ of the data as it randomly splits the data into $H$ subsets and estimates $\hat{T}\_h$ and $\hat{U}\_h$  solely from the subset with index $h$. As a reader, I would be interested in learning whether such a procedure would be required in practice or is merely a tool to facilitate theoretical analysis. When learning from state-action transitions, it is furthermore assumed that the sets $\mathcal{S}^m_{h, s, o}$ (footnote 5) are known in advance. Moreover, the option transition function and utilities are obtained by first learning a one-step model which would suffer from compounding model errors.

I would like to better understand the claim that PEVIO has near-optimal suboptimality. Equation (8) looks like it has an additional factor of $H^2$ under the square root which is quite significant. In practice, $Z^*_\mathcal{O}$ and $\bar{Z}^*_\mathcal{O}$ can be expected to scale with $H$ as well which results in a factor of $H^5$ under the square root which is identical to $VI-LCB$. This would also mean that Corollary 5.3 would not make sense but that it actually depends on the constants whether (8) or the bound for VI-LCB is better. For these reasons I am not quite convinced that the claim “Therefore, Theorems 5.2 and Corollary 5.3 combine to show that [...], options facilitate more sample-efficient learning than primitive actions in both the convergence rate to the optimal value and the actual performance.” is correct with respect to the convergence rate.

Furthermore, the summation in equation (9) seems to effectively introduce another factor of $H$ which is absorbed into a constant. The requirement on the number of episodes in Theorem 5.5 seems to be quite excessive because it scales with $S^9$ which is quite extreme and could never be satisfied in practice.
In general I would expect options to speed things up by a constant factor which is effectively the frequency of option switches. As I do not see why this frequency should automatically decrease on long horizon problems I am not sure if the claim about a faster convergence rate in the conclusion is sufficiently backed.

Typos:
* In line 116 there is a superfluous space before the comma.

**Questions:**

* I would suggest to explicitly spell out in the text that the reward is assumed to be in the range $[0, 1]$ as it is easy to overlook this assumption when it is absorbed into the formal definition of the reward function.
* In Subroutine 3, would it also be possible to learn the option model more directly (without relying on a model for primitive actions), perhaps using importance sampling?

**Limitations:**

A dedicated paragraph or section about limitations is lacking at the moment. I feel like this would help the reader to better place the contributions of the paper. In particular a clear discussion of how the suboptimality bounds for PEVIO compare to each other and to VI-LCB with realistic assumptions on the option switching frequency would be helpful.

---

> ### Author Rebuttal · Authors · 2023-08-03
>
> Thanks for your reviews, we would like to address your concerns as follows.
>
> **1. (Paragraph 1 in Weakness Section) Data Splitting Technique.**
>
> We point out that the data splitting procedure for dataset $\mathcal{D}\_1$ is only to facilitate theoretical analysis. Indeed, by Definition 4.1, a $\xi$-uncertainty quantifier upper bounds the error $|\widehat{U}\_h(s,o)-U_h(s,o)+\sum_{\tau=1}^{H-h+1}\[(\widehat{T}\_h-T_h)\widehat{V}\_{h+\tau}\](s,o)|$. Without splitting the dataset, i.e., we estimate $\widehat{T}\_h$ and $\widehat{V}\_{h+\tau}$ using the same data, it requires to bound concentration terms of the form $(\widehat{T}\_h-T_h)\widehat{V}\_{h+\tau}$. However, after splitting the dataset, as $\widehat{T}\_h$ only depends on $\mathcal{D}\_{1,h}$ and $\widehat{V}_{h+\tau}$ only depends on $\mathcal{D}\_\{1,t>h\}$, $\widehat{T}\_h$ and $\widehat{V}\_{h+\tau}$ are independent. Therefore, we can derive the penalty function as $\Gamma_h(s,o)\leftarrow\widetilde{O}(\frac{H^2}{n_h(s,o)\vee 1})$.
>
> **2. (Paragraph 1 in Weakness Section) Knowledge of $\mathcal{S}_{h,s,o}^m$.**
>
> We point out that while we assume the knowledge of $\mathcal{S}\_{h,s,o}^m$, i.e., the set of state-action pairs that can be reached at timestep $m\ge h$ by using option $o$ at state $s$ and timestep $h$ without terminating, it is only for convenience and simplicity of the proof. Indeed, as it is pointed out in Footnote 5 we can replace $\mathcal{S}\_{h,s,o}^m$ with its superset $\overline{S}\_{h,s,o}^m:=\\{(s,a)\in\mathcal{S}\times\mathcal{A}:\pi_m^o(a|s)>0\\}$, which does not require any prior knowledge on the environment.
>
> **3. (Paragraph 2 in Weakness Section) Near-Optimality of PEVIO.**
>
> We note that in goal-oriented tasks (such an example is given by Fig. 2., Sutton et al., 1999), where the agent aims to reach the goal state as quickly as possible, the options are often designed to minimize the steps of reaching some states from the other. Noting that  $Z_\mathcal{O}^*$ and $\overline{Z}\_\mathcal{O}^*$ only depend on the optimal hierarchical policy, we would have that $Z_\mathcal{O}^*=\overline{Z}\_\mathcal{O}^*=O(1)$, i.e., at the initial state, the learner picks the option that reaches the goal state using minimal steps and does not terminate it until the end. In this example, while the horizon of the problem can be large, i.e., the number of steps to reach the goal state is large, the frequency of options switching is much lower. Similar situations can also be found in tasks involving multiple sub-tasks. For instance, a robot that serves a cup of coffee to a customer can decompose the task into (1) making a cup of coffee, (2) picking up the coffee, (3) going to the table where the customer sits, and (4) serving the coffee. Again, in this example, while it takes a large number of (primitive) actions to finish the task, the need for switching options only emerges after a sub-task is finished. In these learning scenarios, the frequency of option switches, or equivalently, $Z_\mathcal{O}^*$ and $\overline{Z}\_\mathcal{O}^*$, scales with *the number of sub-tasks*, which can be greatly smaller than the horizon $H$ of the problems. Therefore, our results indeed show that options facilitate more sample-efficient learning than primitive actions.
>
> **4. (Paragraph 3 in Weakness Section) Further Clarification of Theorem 5.5.**
>
> We point out that the requirement that $K$ scales with $S^9$ in Theorem 5.5 is merely to simplify the regret bound. As it is stated in the proof of Theorem 5.5 (See Appendix F), the suboptimality bound of *any* $K$ should be $\widetilde{O}(\sqrt{\frac{C\_2\^\textup{option}H^3SZ_\mathcal{O}\overline{Z}\_\mathcal{O}^*}{K}}+\frac{C_2^\textup{option}H^4S^5AO}{K})$, where the second term decays to zero faster than the first term. Hence, when $K$ is sufficiently large, i.e., $K>\widetilde{O}(C_2^\textup{option}H^5S^9A^2O^2/(Z_\mathcal{O}\overline{Z}_\mathcal{O}^*))$, the first term is dominant. Therefore, to simplify the expression in Theorem 5.5, we let $K$ scale with $S^9$.
>
> **5. Typos and (Q1) Range of the Reward.**
>
> We thank the reviewer for pointing out the typo in (L116) and the suggestion of spelling out the range of the reward more explicitly in the text. These will be improved in the later version of the paper.
>
> **6. (Q2) Possible Improvements on Subroutine 3.**
>
> The reviewer asks whether it is possible to learn the option model through importance sampling. We point out that there are two potential problems. First, when options are different, e.g., they select different actions, it is often difficult to have a good estimation of the values using importance sampling. Second, as the importance sampling often takes the form $\frac{\pi_h^o(a|s)}{\pi_h^{o'}(a|s)}$, where $o$ and $o'$ are two options and $\pi$ is the option's policy, the proportion can be arbitrarily large. In this case, the suboptimality bound may be looser than our current results. We agree with the reviewer that techniques like importance sampling could facilitate the implementation of PEVIO in practice and we shall research it in future work.

---

> > ### Comment · Reviewer_V1tj · 2023-08-11
> >
> > Thank you for the response to my questions and concerns.
> >
> > 1. Thank you for the explanation. It makes sense that the splitting technique is used as a tool to simplify the analysis.
> >
> > 2. Ok, I understand that the set $\mathcal{S}^m_{h, s, o}$ is only relevant for the proof.
> >
> > 3. I am not convinced by the example of the option that navigates to the goal in one go. One of the motivating factors for the option framework is temporal abstraction as it facilitates learning in long-horizon tasks. Assuming that a single option can solve the task offloads the task completely to option learning and is not very realistic in more complex environments. Continuing your coffee robot example, if the robot would work a whole shift, then it would have to serve multiple customers. Thus the options for solving the subtasks have to be repeated for every customer and the number of option switches scales with the horizon which should be proportional to the number of customers.
> > I acknowledge that there are scenarios where $Z^*_\mathcal{O}$ and $\bar{Z}^*_\mathcal{O}$ do not scale with $H$. However, in many applications like navigating through a complex environment with options that can only manage short stretches the scaling with $H$ will hold.
> > I would therefore strongly encourage the authors to clarify this implicit assumption (of $Z^*_\mathcal{O}$ and $\bar{Z}^*_\mathcal{O}$ not scaling with $H$ but slower) by explicitly mentioning it before claiming that learning with options has a faster convergence rate. In my opinion this is necessary to prevent misunderstandings.
> >
> > 4. Ok, I can see where the factor $S^9$ is coming from and it is interesting to see the asymptotic behavior. In practical settings the required number of episodes is far too high to be applicable, however. This does not mean that I have a problem with the Theorem, I just wanted to clarify this point.
> >
> > 6. Thank you for the explanation. I can see that importance sampling can complicate the analysis.

---

> > > ### Author Response · Authors · 2023-08-12
> > > **Response to Reviewer V1tj**
> > >
> > > Thank you for your timely reply. Regarding the comments on the coffee robot example (Point 3), we note that the reviewer may confuse between *long-horizon planning* and *repetitions of a task*. That is, serving one customer is a long-horizon planning task, while serving multiple customers is merely repetitions of that task. More importantly, repetitions of a task will *not* increase the sample complexity of learning the optimal hierarchical policy (since if the robot has learned to serve one customer, then she can serve any other customer without additional learning). To be more specific, both $Z_\mathcal{O}^*$ and $\overline{Z}_\mathcal{O}^*$ only depend on the task itself and are irrelevant to the number of times that the task is repeated. We agree with the reviewer that there should be a more thorough and detailed discussion on when learning with options is more sample-efficient than RL with primitive actions. We will improve our writing in the later version of the paper.

---

> > > > ### Comment · Reviewer_V1tj · 2023-08-14
> > > >
> > > > Thank you for your thoughts on the question of how $Z^*_\mathcal{O}$ and $\bar{Z}^*_\mathcal{O}$ scale with the horizon $H$. I do not see a contradiction between subtasks that repeat and long-horizon tasks. In the coffee robot example serving customers might be quite repetitive but other parts of working a shift may not, e.g., cleaning the coffee machine, bringing out the garbage etc. It seems to me that at least some long-horizon tasks would have a combination of subtasks that keep repeating with a fixed frequency (like serving a customer or navigating to the next subgoal when moving around an environment) and some that would not (like closing the shop or dropping a parcel at a target destination).
> > > >
> > > > To mention a very concrete example: Mazes are often used in benchmarks of HRL and scaled up to test the ability of algorithms to deal with longer horizons. If the set of options is kept fixed when doing so and if they are only able to navigate short stretches, then the number of option switches would increase with the horizon $H$ of the maze navigation task.
> > > >
> > > > So as far as I can see how the frequency of option switches scales with $H$  ultimately depends on the concrete scenario. I agree that a more thorough discussion on how the convergence rate of learning with options depends on the horizon $H$ would be a good addition for the paper and I would invite the authors to share their draft for it with the reviewers.

---

> > > > > ### Author Response · Authors · 2023-08-15
> > > > > **A Draft Discussion**
> > > > >
> > > > > Thank you for your follow-up reply. Based on our previous discussion, we provide a draft to the reviewer, which can be added at (L245) of our original paper to provide a more thorough interpretation of the claim that options facilitate more sample-efficient offline learning. We find that this indeed improves the readability of the paper.
> > > > >
> > > > > (L245) ... a higher value than learning with primitive actions. While, in the worst case, both $Z\_\mathcal{O}^*$ and $\overline{Z}\_\mathcal{O}^*$ can scale with $H$ and $HS$, respectively, we note that in many long-horizon planning problems, they often scale with the *number of sub-tasks*, which are greatly smaller, especially for tasks that enable temporal abstraction and the reduction of the state space. For example, although the route-planning task of going from City A to City B by transportation takes thousands of primitive actions to finish, it can be efficiently solved by decomposing into the following sub-tasks: (1) going to the airport/train station in City A; (2) taking transportation to City B; and (3) reaching the final destination in City B, for which options are designed. In this case, since the agent sticks to an option until a sub-task is finished, both $Z\_\mathcal{O}^*$ and $\overline{Z}\_\mathcal{O}^*/S$ may only scale as $o(H)$. In other words, options facilitate more sample-efficient learning through temporal abstraction. Another concrete example is solving a maze, where options are often designed to move agents to bottleneck states (Simsek and Barto, 2004; Solway et al., 2014; Machado et al., 2017) that connect different densely connected regions of the state space, e.g., doorways. In this case, since the agent uses an option until a bottleneck state is reached, the number of states to switch options can be greatly smaller than $S$, i.e., $\overline{Z}_\mathcal{O}^*/H=o(s)$. That is to say, options help improve the sample complexity by the reduction of the state space. Therefore, in these cases, Theorems 5.2 and Corollary 5.3 combine to show that through the temporal abstraction and the reduction of the state space, options facilitate more sample-efficient learning than primitive actions in both the convergence rate to the optimal value and the actual performance.
> > > > >
> > > > > **References**
> > > > >
> > > > > Simsek, O. and Barto, A. G. (2004). Using relative novelty to identify useful temporal abstractions in reinforcement learning. In Proceedings of the Twenty-First International Conference on Machine Learning, ICML ’04, page 95, New York, NY, USA. Association for Computing Machinery.
> > > > >
> > > > > Solway, A., Diuk, C., Córdova, N., Yee, D., Barto, A. G., Niv, Y., and Botvinick, M. M. (2014). Optimal behavioral hierarchy. PLOS Computational Biology, 10(8):1–10.
> > > > >
> > > > > Machado, M. C., Bellemare, M. G., and Bowling, M. (2017). A Laplacian framework for option discovery in reinforcement learning. In Precup, D. and Teh, Y. W., editors, Proceedings of the 34th International Conference on Machine Learning, volume 70 of Proceedings of Machine Learning Research, pages 2295–2304. PMLR.

---

> > > > > > ### Comment · Reviewer_V1tj · 2023-08-15
> > > > > >
> > > > > > Thank you for providing a detailed discussion of the scaling of $Z^*_\mathcal{O}$ and $\bar{Z}^*_\mathcal{O}$ with $H$ and $S$.
> > > > > >
> > > > > > I think this makes it easier for the reader to place Theorem 5.2 and Corollary 5.3 and to understand the conclusions you draw from them.
> > > > > >
> > > > > > I appreciate this addition to the paper and have raised my score.

---

> > > > > > > ### Author Response · Authors · 2023-08-15
> > > > > > >
> > > > > > > Thank you very much for raising your score!

---

### Official Review · Reviewer_FY62 · 2023-07-10

**Soundness:** 3 good
**Presentation:** 3 good
**Contribution:** 2 fair
**Rating:** 6
**Confidence:** 2

**Summary:**

In this paper, the authors analyze the sample complexity for offline reinforcement learning with options. In particular, the sample complexity is measured by the suboptimality (the shortfall in the value function) of a hierarchical policy compared to the optimal hierarchical policy. The authors propose the pessimistic Value Iteration for Learning with Options algorithm and propose bounds when using two popular data collection methods: one collecting state-option-utility tuples and one collecting state-action-reward tuples. The paper shows that using options can lead to a faster convergence rate (to the value of the optimal hierarchical policy over given options).

**Strengths:**

- The paper investigates an important problem in the HRL literature: how does using options impact the efficiency of offline learning algorithms, considering different data collection approaches.
- The paper is well-written, drawing parallels with existing work whenever possible.
- The paper is very thorough with the mathematical formulation, assumptions, and definitions.

**Weaknesses:**

The theory is based on the assumption that a set of options is already available, and it focuses on the suboptimality of the hierarchical policy over those options (when compared to the optimal hierarchical policy over those options). Thus, the overall performance of the final hierarchical policy is dependent on the quality of the underlying options (as their policy is not being updated). Also, the intuition behind some of the other assumptions necessary to prove the results presented in the paper could be improved (see questions below).

**Questions:**

- Could the authors please provide more details for how $\theta_h^\mu(s,o)$ relates to the policy $\mu$?
- There is possibly a type in line 125, $(h, s) \in [H] \times \mathcal O$.
- Theorem 3 holds “when dataset $\mathcal D$ sufficiently covers the trajectories induced by $\mu^*$”. Could the authors provide some intuition for how often this holds, or how one could verify if it holds?
- Could the authors provide more details about the penalty function $\Gamma$? For instance, what is it’s intuitive meaning, why does it have the format described in Subroutines 2 and 3?
- Theorem 4.2 holds if the penalty function is an $\xi$-uncertainty quantifier. Could the authors provide some intuition for how often this holds, or how one could verify if it holds?
- How strong are assumptions 5.1 and 5.4? How could one ensure that they are satisfied?
- How does the magnitude of $C^*$ compares with $C^{option}$? How does the magnitude of these constants generally compare with the other terms in the bounds?

**Limitations:**

The authors discuss the limitations of the work. No major concerns.

---

> ### Author Rebuttal · Authors · 2023-08-03
>
> Thanks for your reviews, we would like to address your concerns as follows.
>
> **(Q1) Explanation of $\theta_h^\mu(s,o)$.**
>
> By its definition in (L121), $\theta_h^\mu(s,o)$ is the probability of visiting the state-option pair $(s,o)$ at timestep $h$, when following the hierarchical policy $\mu$. For example, let $\mu$ and $\mu'$ denote two hierarchical policies. Particularly, in the first step, we have that $\mu_1(o|s_1)=1$ and $\mu'_1(o|s_1)=0$ for some option $o$, i.e., $\mu$ *always* picks option $o$ while $\mu'$ *never* picks option $o$. In this case, we have that $\theta_1^\mu(s_1,o)=1$ while $\theta_1^{\mu'}(s_1,o)=0$.
>
> **(Q2) Typo in (L125).**
>
> The expression in (L125) should be $(h,s)\in[H]\times\mathcal{S}$. We thank the reviewer for pointing out the typo.
>
> **(Q3) Sufficient Coverage in Theorem 3.**
>
> As it is pointed out in (L144) the sufficient coverage assumption in Theorem 3 refers to the condition that $\max_{h,s,o}\frac{\theta^{\mu^*}_h(s,o)}{\theta^\rho_h(s,o)}\le C^\text{option}<\infty$, where $\mu^*$ is the optimal hierarchical policy, $\rho$ is the behavior policy to collect the dataset, and $\theta$ is the probability of visiting the state-option pairs. We note that one case that this assumption holds is the behavior policy $\rho$ is uniformly random.
>
> **(Q4) Intuition of the Penalty Function and Subroutines.**
>
> The meaning of the penalty function $\Gamma$ can be directly seen from Inequality (5) in Definition 4.1, which says that with high probability, $\Gamma$ should bound the error in estimating the $Q$-function for each $(h,s,o)\in[H]\times\mathcal{S}\times\mathcal{O}$. To further interpret Inequality (5), first note that the $Q$-function (See Equation (1)) is a linear combination of the utility $U_h$ and the value $V_{h+\tau}$ in future steps. Hence, the error of estimated $Q$-function can be further decomposed as the error in estimating the utility, i.e., $\widehat{U}\_h-U_h$, and the error in estimating the future values, i.e., $(\widehat{T}\_h-T_h)\widehat{V}_{h+\tau}$, which correspond to the two terms in Inequality (5) exactly. The reviewer also asks why different formats in Subroutines 2 and 3 are given to compute $\Gamma$. We point out that this is because that when different datasets are provided, the computations for $\Gamma$ are also different. For example, when $(s,o,u)$ (i.e., state-option-utility) tuples are given (Dataset $\mathcal{D}_1$), the computation of $\Gamma_h(s,o)$ in Line 8 of Subroutine 2 needs to count the number of visits to $(h,s,o)$ in the dataset. However, the same computation cannot be done when only $(s,a,r)$ (i.e., state-action-reward tuples) are given (Dataset 2). Therefore, we need to design a different subroutine to compute $\Gamma$ in Subroutine 3.
>
> **(Q5) Verification of $\xi$-uncertainty Quantifier.**
>
> The reviewer asks when the penalty function is an $\xi$-uncertainty quantifier. We point out that by our designs of $\Gamma$, i.e., Line 8 in Subroutine 2 and Line 18 in Subroutine 3, $\Gamma$ is already an $\xi$-uncertainty quantifier, without *any* further conditions or assumptions. The detailed proof can be found in Lemmas E.1 and F.2 in the Appendices.
>
> **(Q6) Assumptions 5.1 and 5.4.**
>
> The reviewer asks how strong are Assumptions 5.1 and 5.4. We note that they immediately hold when the behavior policy collecting the dataset is uniformly random, which is quite common in practice. In addition, we point out that they are standard in the offline RL literature (Rashidinejad et al., 2021; Jin et al., 2021; Xie et al., 2021). As a comparison, there is another line of studies on offline RL that assumes full coverage of the behavior policy [1, 2], i.e., $\min_{h,s,a}d_h^\rho(s,a)>0$, which is much more strict than Assumptions 5.1 and 5.4.
>
> **(Q7) Comparison of $C^\*$ and $C^\{\text{option}}$.**
>
> In general, $C^*$ and $C^\textup{option}$ are incomparable as they depend on the design of options, the behavior policy to collect the dataset, and the transition kernel of the MDP. Nonetheless, we show that (the upper bound of) $C\^\textup{option}\_1$ can be smaller than (the upper bound of) $C^*$ in the following example. Assume that we have access to a generative model that given any $(h,s,o)$, returns the timestep-state pair $(h+\tau,s')$ that option $o$ is terminated. In this case, if we uniformly sample state-option pair to collect the dataset, i.e., $(s,o)\sim\textup{Uniform}(\mathcal{S}\times\mathcal{O})$, for each timestep $h$, then we have that $C^\textup{option}\_1=\max_{h,s,o}\frac{\theta^{\mu^*}_h(s,o)}{\theta^\rho_h(s,o)}=\max\_{h,s,o}\frac{\theta^{\mu^*}_h(s,o)}{1/SO}\le SO$, where $\mu^*$ is the optimal hierarchical policy and $O$ is the number of options. In addition, in RL with primitive actions, if we have access to a generative model that given any $(s,a)$, returns $s'$ in the next timestep and we still uniformly sample $(s,a)\in\mathcal{S}\times\mathcal{A}$ for each $h$, then we have that $C^\*=\max\_{h,s,a}\frac{d^{\pi^*}\_h(s,a)}{d^\rho\_h(s,a)}=\max\_{h,s,a}\frac{d^{\pi^*}\_h(s,a)}{1/SA}\le SA$, where $\pi^*$ is the optimal policy defined on primitive actions and $A$ is the number of actions. Since $O$ is often smaller than $A$, the upper bound for $C\^\textup{option}_1$ is smaller than the one for $C^*$.
>
> **References**
> 1. Ming Yin, Yu Bai, Yu-Xiang Wang (2021). Near-Optimal Offline Reinforcement Learning via Double Variance Reduction.
> 2. Ming Yin, Yu Bai, Yu-Xiang Wang (2021). Near-Optimal Provable Uniform Convergence in Offline Policy Evaluation for Reinforcement Learning.

---

> > ### Comment · Reviewer_FY62 · 2023-08-21
> >
> > Thank you for your thorough response. After reading it, and reading the other reviews and discussion, I increased my score. Thank you for clarifying my questions!

---

> > > ### Author Response · Authors · 2023-08-22
> > >
> > > Thank you very much for raising your score!

---

### Official Review · Reviewer_UEUz · 2023-07-21

**Soundness:** 3 good
**Presentation:** 2 fair
**Contribution:** 3 good
**Rating:** 6
**Confidence:** 2

**Summary:**

The authors provide a sample complexity analysis of offline RL with options. A sub-optimality analysis was performed considering two data collection schemes, (D1) state-option-utility collection and (D2) state-action-reward collection for the proposed pessimistic value iteration algorithm (PEVIO). The analysis reveals that (D1) is storage efficient and has a fast convergence rate, while (D2) has weak assumptions and allows flexible evaluation of new options.

**Strengths:**

(S1) Solid theoretical analysis

(S2) Analysis of the Pros and cons of the data-collection procedures

**Weaknesses:**

(W1) Scope of the theory is not clearly stated.

(W2) Background and discussion on key assumptions seem insufficient.

(W3) Unclear parameter update equation in the proposed Algorithm.


**Questions:**

I think this study is interesting since it draws conclusions about the advantages and disadvantages of the data collection schemes through solid theoretical analysis. However, as mentioned above, it appears the current version of this paper has three weaknesses. I would appreciate it if the authors could answer the following questions.

-Is the scope of the authors' theory and algorithms limited to Tabular MDPs (MDPs with discrete state and action space)? I can guess this from the content, but some clear description is needed.
-Are there any related works and background on the finite concentrability assumption (Assumption 5.1 and 5.4), which is the central assumption of the analysis? It is helpful if list papers that adopt the same or similar assumptions and/or discuss the approach with different assumptions, etc. That will help to place this study in the context of the research.
-How to update the penalty function in the Proposed PEVIO algorithm? It only provides the order-level update equation, unlike the (original) PEVI. Please give us some guidelines for implementation even if it is through numerical experiments.

Minor Comment:
-Citation needed at (L147) and (L217)

**Limitations:**

No "Limitations" section is provided by the author.

---

> ### Author Rebuttal · Authors · 2023-08-03
>
> Thanks for your reviews, we would like to address your concerns as follows.
>
> **(Q1) Scope of the Theory.**
>
> The results in this paper are limited to tabular MDPs. We will clearly illustrate this assumption in the preliminaries section in the later version of the paper.
>
> **(Q2) Finite Concentrability Assumption and Related Works.**
>
> As we have illustrated in the related work section, the finite single-policy concentrability assumption is quite standard and is adopted by the works of Rashidinejad et al. (Definition 1, 2021), Jin et al. (Corollary 4.5, 2021), and Xie et al. (Assumption A, 2021). We will improve our expression in the later version of the paper.
>
> **(Q3) Update of the Penalty Function.**
>
> We point out that to update the penalty function, the algorithm only needs to count the visits to state-action or state-option pairs. To see this, we provide the exact expression of the penalty function $\Gamma$ as follows. When state-option-utility tuples are provided (Dataset $\mathcal{D}\_1$), the penalty in Line 8 of Subroutine 2 is computed by $\Gamma_h(s,o)\leftarrow2\sqrt{\frac{H^2}{n_h(s,o)\vee 1}\log(\frac{2HSO}{\xi})}$, where $n_h(s,o)$ is the number of visits to $(h,s,o)$ in subdataset $\mathcal{D}\_{1,h}$ and $\xi\in(0,1)$ is the probability of failure. In addition, when state-action-reward tuples are provided (Dataset $\mathcal{D}\_2$), the penalty in Line 18 of Subroutine 18 is computed by $\Gamma_h(s,o)\leftarrow2\sqrt{\sum_{m=h}^H\sum_{(s,a)\in\mathcal{S}_{h,s,o}^m}\frac{HS}{N_m(s,a)\vee 1}\log(\frac{2HSA}{\xi})}+2\cdot\phi_h(s,o)\log(\frac{2HSA}{\xi})$, where $N_h(s,a)$ is the number of visits to $(h,s,a)$ in $\mathcal{D}_2$ and $\phi_h$ is defined in Equation (32) in the Appendices.
>
> **(Q4) Citations.**
>
> The citation at (L147) should be (Xie et al., 2021) and the citation at (L217) should be (Rashidinejad et al., 2021; Jin et al., 2021; Xie et al., 2021).

---

> > ### Comment · Reviewer_UEUz · 2023-08-15
> >
> > Thank you for answering my questions and concerns.
> > My main concern was about Q3 and your answer helped me understand that it is not a problem.
> > Therefore, I have raised my score for this paper.

---

> > > ### Author Response · Authors · 2023-08-17
> > >
> > > Thank you very much for raising your score!

---

### Official Review · Reviewer_BJAs · 2023-07-26

**Soundness:** 3 good
**Presentation:** 2 fair
**Contribution:** 2 fair
**Rating:** 7
**Confidence:** 3

**Summary:**

This analysis is the study of sample complexity in offline reinforcement learning (RL) with options, highlighting potential risks in online environmental exploration. The paper proposes the innovative PEssimistic Value Iteration for Learning with Options (PEVIO) algorithm and provides near-optimal suboptimality bounds for two common data-collection methods. The research underscores PEVIO's superior performance and faster convergence when optimally designed options are used or offline data is limited. Moreover, the study explores two different data collection methods, providing unique insights into their respective benefits and challenges. Overall, this research offers valuable contributions to the practical implementation and understanding of offline RL with options.

**Strengths:**

This paper tries to put the innovative exploration of sample complexity in offline reinforcement learning (RL) with options, introducing the novel PEVIO algorithm. Combining  theory with sound empirical evaluations provides robust insights into different data collection methods and their implications. Despite the complexity of the subject, the paper maintains fair clarity. Its practical significance is clear, as it tries to fill a literature gap and holds the potential to influence future RL system design, making it a vital addition to the field of RL.

**Weaknesses:**

While the paper is undoubtedly insightful, further exploration of the influence of option complexity on PEVIO's effectiveness could enhance its contribution. Expanding the experimental setup to include varied environments and offline datasets might be beneficial to validate PEVIO's performance robustly. I think a more thorough discussion of the trade-offs between the two data collection methods could aid practitioners in making more informed choices in diverse application contexts.

**Questions:**

I just want to understand on curiosity - Is there any thought that the work may be helpful or this approach can also be helpful in  Hybrid (Online +Offline RL). Any substantial logic for this?
Also, Can the authors discuss potential extensions or improvements of the PEVIO algorithm?

**Limitations:**

Author mentioned the limitations in some way Pros and Cons. So, OK. But I would love to understand explicitly about the scope limitation of the algorithm in more detail if there is any additional material.

---

> ### Author Rebuttal · Authors · 2023-08-03
>
> Thanks for your reviews, we would like to address your concerns as follows.
>
> **(Q1) Connecting to Hybrid RL.**
>
> The reviewer asks whether our results can also facilitate hybrid RL. While this is out of the scope of the paper, we would like to provide some comments as follows. In hybrid RL, it is often the case that a policy is pre-trained or the $Q$-function is estimated in the offline phase using a batch dataset and then fine-tuned in the online phase. Hence, the performance during online learning is often affected by, e.g., the quality of the pre-trained policy or estimated values. As our paper shows that offline learning with options is more sample-efficient than RL with primitive actions, i.e., the output policy enjoys lower suboptimality, it may also help the learning in hybrid RL. For example, one possible way is to use options to obtain better estimated values in the offline phase, which further improves online performance. Therefore, we believe that the results presented in this paper are of importance for not only the understanding of the options framework in offline RL but could also provide insights into relevant research fields.
>
> **(Q2) Limitations and Potential Improvements of PEVIO.**
>
> There could be several improvements to PEVIO. For example, our paper primarily focuses on tabular MDPs (finite state space). Hence, one direction is to extend PEVIO to more general MDPs (infinite state space), e.g., linear MDPs. In addition, when state-action-reward tuples are provided (Dataset $\mathcal{D}_2$), the algorithm needs to estimate the one-step transition model to compute the $Q$-function, which can be computationally inefficient. Therefore, reducing the computation of PEVIO remains an interesting future work.

---

### Decision · Program_Chairs · 2023-09-21

**Decision:**

Accept (poster)

**Comment:**

The paper examines offline RL with options and studies the effect of having options on the performance of the offline RL algorithm. They also analyse the effects of two different data collection strategies. The paper is well-executed and well-written. In terms of shortcomings, the paper: (1) has a limited experimental evaluation (2) treats table-lookup problems only (3) assumes that set of options is given (as opposed to learned) (4) provides an algorithm quite far from a practical larger-scale implementation. Overall I think these are OK in a theoretical paper (and the reviewers seem to agree).